

# Experimental assessment of the sensitivity of an estuarine phytoplankton fall bloom to acidification and warming

Robin Bénard[1], Maurice Levasseur[1], Michael Grant Scarratt[2], Marie-Amélie Blais[1], Alfonso Mucci[3], Gustavo Ferreyra[4], Michel Starr[2], Michel Gosselin[4], Jean-Éric Tremblay[1], Martine Lizotte[1]

[1]Département de biologie, Université Laval, 1045 avenue de la Médecine, Québec, Québec G1V 0A6, Canada
[2]Fisheries and Oceans Canada, Maurice Lamontagne Institute, P.O. Box 1000, Mont-Joli, Québec G5H 3Z4, Canada
[3]Department of Earth and Planetary Sciences, McGill University, 3450 University Street, Montréal, Québec H3A 2A7, Canada
[4]Institut des sciences de la mer de Rimouski (ISMER), Université du Québec à Rimouski, 310 allée des Ursulines, Rimouski, Québec G5L 3A1, Canada

*Correspondence*: Robin Bénard (robin.benard.1@ulaval.ca)

**Abstract.** We investigated the combined effect of ocean acidification and warming on the dynamics of the phytoplankton fall bloom in the Lower St. Lawrence Estuary (LSLE), Canada. Twelve 2600 L mesocosms were set to initially cover a wide range of $pH_T$ (pH on the total proton scale) from 8.0 to 7.2 corresponding to a range of $pCO_2$ from 440 to 2900 µatm, and two temperatures (in situ and +5 °C). The 13-day experiment captured the development and decline of a nanophytoplankton bloom dominated by the chain-forming diatom *Skeletonema costatum*. During the development phase of the bloom, increasing $pCO_2$ influenced neither the magnitude nor the net growth rate of the nanophytoplankton bloom whereas increasing the temperature by 5 °C stimulated the chlorophyll *a* (Chl *a*) growth rate and particulate primary production ($P_P$) by 50 % and 160 %, respectively. During the declining phase of the bloom, warming accelerated the loss of diatom cells and negatively affected $P_P$. Due to the countervailing responses of the plankton community to warming during the two phases of the experiment, the time-integrated primary production was not significantly affected over the full duration of the study. The diatom bloom was paralleled by a gradual decrease in the abundance of photosynthetic picoeukaryotes and followed by a bloom of picocyanobacteria. Increasing $pCO_2$ and warming did not influence the abundance of picoeukaryotes, but warming stimulated picocyanobacteria proliferation. Overall, our results suggest that warming, rather than acidification, is more likely to alter phytoplankton autumnal bloom development in the LSLE in the decades to come. Future studies examining a broader gradient of temperatures should be conducted over a larger seasonal window in order to better constrain the potential effect of warming on the development of blooms in the LSLE and its impact on the fate of primary production.

## 1. Introduction

Anthropogenic emissions have increased atmospheric carbon dioxide ($CO_2$) concentrations from their pre-industrial value of 280 to 412 ppm in 2017, and concentrations of 850–1370 ppm are expected by the end of the century under the business-as-usual scenario RCP 8.5 (IPCC, 2013). The global ocean has already absorbed about 28 % of these anthropogenic $CO_2$ emissions (Le Quéré et al., 2015), leading to a global pH decrease of 0.11 units (Gattuso et al., 2015), a phenomenon known



as Ocean Acidification (OA). The surface ocean pH is expected to decrease by an additional 0.3–0.4 units under the RCP 8.5 scenario by 2100, and as much as 0.8 units by 2300 (Caldeira and Wickett, 2005; Doney et al., 2009; Feely et al., 2009). The accumulation of anthropogenic $CO_2$ in the atmosphere also results in an increase in the Earth's heat content that is primarily absorbed by the ocean (Wijffels et al., 2016), leading to an expected rise of sea surface temperatures of 3 to 5 °C by 2100 (IPCC, 2013). Whereas the effect of increasing atmospheric $CO_2$ partial pressures ($pCO_2$) on ocean chemistry is relatively well documented, the potential impacts of OA on marine organisms and how their response to OA will be modulated by the concurrent warming of the ocean surface waters are still the subject of much debate (Boyd and Hutchins, 2012; Gattuso et al., 2013).

Over the last decade, there has been increasing interest in the potential effects of OA on marine organisms (Kroeker et al., 2013). The first experiments were primarily conducted on single phytoplankton species (reviewed in Riebesell and Tortell, 2011), but subsequent mesocosm experiments highlighted the impact of OA on the structure and productivity of complex plankton assemblages (Riebesell et al., 2007, 2013). Due to their widely different initial and experimental conditions, these ecosystem-level experiments generated contrasting results (Schulz et al., 2017) but some general patterns nevertheless emerged. For example, diatoms generally benefit from higher $pCO_2$ through stimulated photosynthesis and growth rates since the increase in $CO_2$ concentrations compensates for the low affinity of RubisCO towards $CO_2$ (Giordano et al., 2005; Gao and Campbell, 2014). Although most phytoplankton species have developed carbon concentration mechanisms (CCM) to compensate for the low affinity of RubisCO towards $CO_2$, CCM efficiencies differ between taxa, rendering predictions of the impact of a $CO_2$ rise on the downregulation of CCM rather difficult (Raven et al., 2014). For example, some studies unexpectedly reported no significant or very modest stimulation of primary production under elevated $CO_2$ concentrations (Engel et al., 2005; Eberlein et al., 2017). OA can ultimately affect the structure of phytoplankton assemblages. Small cells such as photosynthetic picoeukaryotes can benefit directly from an increase in $pCO_2$ as $CO_2$ can passively diffuse through their boundary layer (Beardall et al., 2014), and the smallest organisms within this group could benefit most from the increase (Brussaard et al., 2013). Accordingly, OA experiments have typically favoured smaller phytoplankton cells (Yoshimura et al., 2010; Brussaard et al., 2013; Morán et al., 2015), although the proliferation of larger cells has also been reported (Tortell et al., 2002). Hence, generic predictions of phytoplankton community responses to OA are challenging.

Few recent studies have investigated the combined effects of OA and warming on natural phytoplankton assemblages (Hare et al., 2007; Feng et al., 2009; Maugendre et al., 2015; Paul et al., 2015, 2016). Laboratory experiments have shown that OA and warming could together increase photosynthetic rates, but at the expense of species richness, the reduction of diversity predominantly imputable to warming (Tatters et al., 2013). Results of an experiment conducted with a natural planktonic community from the Mediterranean Sea showed no effect of a combined warming and decrease in pH on primary production, but higher picocyanobacteria abundances were observed in the warmer treatment (Maugendre et al., 2015). Shipboard microcosm incubations conducted in the northern South China Sea displayed higher phytoplankton biomass, daytime primary productivity and dark community respiration under warmer conditions, but these positive responses were cancelled at low pH (Gao et al., 2017). In contrast, a mesocosm experiment carried out with a fall planktonic community from the western Baltic



Sea led to a decrease in phytoplankton biomass under warming, but combined warming and increased $pCO_2$ led to an increase in biomass (Sommer et al., 2015). Results from experiments where the impacts of $pCO_2$ and temperature are investigated individually may be misleading as multiple stressors can interact antagonistically or synergistically, sometimes in a nonlinear, unpredictable fashion (Todgham and Stillman, 2013; Boyd et al., 2015; Riebesell and Gattuso, 2015; Gunderson et al., 2016). The Lower St. Lawrence Estuary (LSLE) is a large (9350 km$^2$) segment of the greater St. Lawrence Estuary (d'Anglejan, 1990). From June to September, the LSLE is characterized by a dynamic succession in the phytoplankton community, mostly driven by changes in light and nutrient availability through variations in the intensity of vertical mixing (Levasseur et al., 1984). The spring and fall blooms are mostly comprised of diatoms, with simultaneous nitrate and silicic acid exhaustion ultimately limiting primary production (Levasseur et al., 1987; Roy et al., 1996). How OA and warming may affect these blooms and primary production has never been investigated in the LSLE. The OA problem is complex in estuarine and coastal waters where freshwater runoff, tidal mixing, and high biological activity contribute to variations in $pCO_2$ and pH on different time scales (Duarte et al., 2013). The surface mixed-layer $pCO_2$ in the LSLE varies spatially from 139 to 548 µatm and is strongly modulated by biological productivity (Dinauer and Mucci, 2017). Surface $pH_T$ has been shown to vary from 7.85 to 7.93 in a single tidal cycle in the LSLE, nearly as much as the world's oceans have experienced in response to anthropogenic $CO_2$ uptake over the last century (Caldeira and Wickett, 2005; Mucci et al., 2017).

The main objective of this study was to experimentally assess the sensitivity of the LSLE phytoplankton fall assemblage to a large $pCO_2$ gradient at two temperatures (in situ and +5 °C). Whether lower trophic-level microorganisms thriving in a highly variable environment will show higher resistance or resilience to future anthropogenic forcings is still a matter of speculation.

## 2. Material and methods

### 2.1 Mesocosm setup

The mesocosm system consists of two thermostated full-size ship containers each holding six 2600 L mesocosms (Aquabiotech®). The mesocosms are cylindrical with a cone-shaped bottom within which mixing is achieved using a propeller fixed near the top. Each enclosure is sealed with a Plexiglas cover allowing the transmission of 90 % of photosynthetically active radiation (PAR; 400–700 nm), 50–85 % of solar UVB (280–315 nm) and 85–90 % of UVA (315–400 nm). The mesocosms are equipped with individual, independent temperature probes (AQBT-Temperature sensor, accuracy ± 0.2 °C). Temperature in the mesocosms was measured every 15 minutes during the experiment, and the control system triggered either a resistance heater (Process Technology TTA1.8215) located near the middle of the mesocosm or a pump-activated glycol refrigeration system to maintain the set temperature. The pH in each mesocosm was monitored every 15 minutes using Hach® PD1P1 probes (± 0.02 pH units) connected to Hach® SC200 controllers, and positive deviations from the target values activated peristaltic pumps linked to a reservoir of artificial seawater equilibrated with pure $CO_2$ prior to the onset of the experiment. This system maintained the pH of the seawater in the mesocosms within ± 0.02 pH units of the targeted values by lowering



the pH during autotrophic growth but could not increase the pH during bloom senescence when the $pCO_2$ rose and pH
decreased.

## 2.2 Setting

The water was collected at 5 m depth near Rimouski harbour (48° 28' 39.9" N, 68° 31' 03.0" W) on the 27th of September 2014.
In situ conditions were: salinity = 26.52, temperature = 10 °C, nitrate $(NO_3^-)$ = 12.8 ± 0.6 µmol $L^{-1}$, silicic acid
$(Si(OH)_4)$ = 16 ± 2 µmol $L^{-1}$, and soluble reactive phosphate (SRP) = 1.4 ± 0.3 µmol $L^{-1}$. The same day (indicated as day -5
hereafter), the water was filtered through a 250 µm mesh while simultaneously filling the 12 mesocosm tanks by gravity with
a custom made 'octopus' tubing system. The initial in situ temperature of 10 °C was maintained in the twelve mesocosms for
the first 24 h (day -4). After that period, the six mesocosms in one container were maintained at 10 °C while temperature was
gradually increased to 15 °C during day -3 in the six mesocosms of the other container. To avoid subjecting the planktonic
communities to excessive stress due to sudden changes in temperature and pH while setting the experiment, the mesocosms
were left to acclimatize on day -2 before acidification was carried out over day -1. One mesocosm from each temperature-
controlled container was not pH-controlled to assess the community response to the freely fluctuating pH. These two
mesocosms were labelled "Controls" as the initial in situ pH was allowed to fluctuate over time with the development of the
phytoplankton bloom. The other mesocosms were set to cover a range of $pH_T$ of ca. 8.0 to ca. 7.2 corresponding to a $pCO_2$
gradient of ca. 440 to ca. 2900 µatm after acidification was carried out.

## 2.3 Seawater analysis

The mesocosms were sampled between 05:00 and 08:00 a.m. every day. Seawater for carbonate chemistry, nutrients, and
primary production were collected directly from the mesocosms as close to sunrise as possible. Seawater was also collected in
20 L carboys for the determination of chlorophyll *a* (Chl *a*), taxonomy, and other variables. Samples for salinity were taken
from the artificial seawater tanks and in the mesocosms on day -3, 3 and 13. The samples were collected in 250 mL plastic
bottles and stored in the dark until analysis was performed using a Guildline Autosal 8400B Salinometer during the following
months.

### 2.3.1 Carbonate chemistry

Carbonate chemistry parameters were determined using methods described in Mucci et al. (2017). Briefly, water samples for
pH (every day) and total alkalinity (TA, every 3–4 days) measurements were, respectively, transferred from the mesocosms to
125 mL plastic bottles without headspace and 250 mL glass bottles. A few crystals of $HgCl_2$ were added to the glass bottles
before sealing them with a ground-glass stopper and Apiezon® Type-M high-vacuum grease. The pH was determined within
hours of collection, after thermal equilibration at 25.0 ± 0.1 °C, using a Hewlett-Packard UV-Visible diode array
spectrophotometer (HP-8453A) and a 5 cm quartz cell with phenol red (PR; Robert-Baldo et al., 1985) and *m*-cresol purple
(mCP; Clayton and Byrne, 1993) as indicators. Measurements were carried out at the wavelength of maximum absorbance of





the protonated (HL) and deprotonated (L) indicators. Comparable measurements were carried out using a TRIS buffer prepared
at a practical salinity of ca. 25 before and after each set of daily measurements (Millero, 1986).
The pH on the total proton concentration scale ($pH_T$) of the buffer solutions and samples at 25 °C was calculated according to
the equation of Byrne (1987), using the salinity of each sample and the $HSO_4^-$ association constants given by Dickson (1990).
The TA was determined on site within one day of sampling by open-cell automated potentiometric titration (Titrilab 865,
Radiometer®) with a pH combination electrode (pHC2001, Red Rod®) and a dilute (0.025N) HCl titrant solution. The titrant
was calibrated using Certified Reference Materials (CRM Batch#94, provided by A. G. Dickson, Scripps Institute of
Oceanography, La Jolla, USA). The average relative error, based on the average relative standard deviation on replicate
standard and sample analyses, was better than 0.15 %. The carbonate chemistry parameters at in situ temperature were then
calculated using the computed $pH_T$ at 25 °C in combination with the measured TA using $CO_2SYS$ (Pierrot et al., 2006) and
the carbonic acid dissociation constants of Cai and Wang (1998).

### 2.3.3 Nutrients

Samples for $NO_3^-$, $Si(OH)_4$, and SRP analyses were collected directly from the mesocosms every day, filtered through
Whatman GF/F filters and stored at -20 °C in acid washed polyethylene tubes until analysis by a Bran and Luebbe Autoanalyzer
III using the colorimetric methods described by Hansen and Koroleff (2007). The analytical detection limit was 0.03 µmol $L^{-1}$
for $NO_3^-$ plus nitrite ($NO_2^-$), 0.02 µmol $L^{-1}$ for $NO_2^-$, 0.1 µmol $L^{-1}$ for $Si(OH)_4$, and 0.05 µmol $L^{-1}$ for SRP.

### 2.3.4 Plankton biomass, composition and enumeration

Duplicate subsamples (100 mL) for Chl *a* determination were filtered onto Whatman GF/F filters. Chl *a* concentrations were
measured using a 10-AU Turner Designs fluorometer, following a 24 h extraction in 90 % acetone at 4 °C in the dark without
grinding (acidification method: Parsons et al., 1984). The analytical detection limit for Chl *a* was 0.05 µg $L^{-1}$.
Pico- (0.2–2 µm) and nanophytoplankton (2–20 µm) cell abundances were determined daily by flow cytometry. Sterile
cryogenic polypropylene vials were filled with 4.95 mL of seawater to which 50 µL of glutaraldehyde Grade I (final
concentration = 0.1 %, Sigma Aldrich; Marie et al., 2005) were added. Duplicate samples were flash frozen in liquid nitrogen
after standing 15 minutes at room temperature in the dark. These samples were then stored at -80 °C until analysis. After
thawing to ambient temperature, samples were analyzed using a FACS Calibur flow cytometer (Becton Dickinson) equipped
with a 488 nm argon laser. The abundances of nanophytoplankton and picophytoplankton, which includes photosynthetic
picoeukaryotes and picocyanobacteria, were determined by their autofluorescence characteristics and size (Marie et al., 2005).
The biomass accumulation and nanophytoplankton growth rates were calculated by the following equation:
$\mu = \ln (N_2/N_1) / (t_2 - t_1),$                      (1)
where $N_1$ and $N_2$ are the biomass or cell concentrations at given times $t_1$ and $t_2$, respectively.



Microscopic identification and enumeration for eukaryotic cells larger than 2 µm was conducted on samples taken from each
mesocosm on three days: day -4, the day when maximum Chl *a* was attained in each mesocosm, and day 13. Samples of
250 mL were collected and preserved with acidic Lugol solution (Parsons et al., 1984), then stored in the dark until analysis.
Cell identification was carried out at the lowest possible taxonomic rank using an inverted microscope (Zeiss Axiovert 10) in
accordance with Lund et al. (1958). The main taxonomic references used to identify the phytoplankton were Tomas (1997)
and Bérard-Therriault et al. (1999).

**2.3.5 Primary production**

Primary production was determined daily using the $^{14}$C-fixation incubation method (Knap et al., 1996; Ferland et al., 2011).
One clear and one dark 250 mL polycarbonate bottle were filled from each mesocosm at dawn and spiked with 250 µL of
NaH$^{14}$CO$_3$ (80 µCi mL$^{-1}$). One hundred µL of 3-(3,4-dichlorophenyl)-1,1-dimethylurea (DCMU) (0.02 mol L$^{-1}$) was added to
the dark bottles to prevent active fixation of $^{14}$C by phytoplankton (Legendre et al., 1983). The total amount of radioisotope in
each bottle was determined by immediately pipetting 50 µL subsamples into a 20 mL scintillation vial containing 10 mL of
scintillation cocktail (Ecolume$^{TM}$) and 50 µL of ethanolamine (Sigma). Bottles were placed in separate incubators, at either
10 °C or 15 °C, under reduced (30 %) natural light for 24 h, which corresponds to the light transmittance at mid-mesocosm
depth.
At the end of the incubation periods, 3 mL were transferred to a scintillation vial for determination of the total primary
production (P$_T$), 3 mL were filtered through a syringe filter (GD/X 0.7 µm) to estimate daily photosynthetic carbon fixation
released in the dissolved organic carbon pool (P$_D$). The remaining volume was filtered onto a Whatman GF/F filter to measure
the particulate primary production (P$_P$). Vials containing the P$_T$ and P$_D$ samples were acidified with 500 µL of HCl 6 N, allowed
to sit for 3 h under a fume hood, then neutralized with 500 µL of NaOH 6 N. The vials containing the filters were acidified
with 100 µL of 0.05 N HCl and left to fume for 12 h. Fifteen mL of scintillation cocktail were added to the vials and they were
stored pending analysis using a Tri-Carb 4910TR liquid scintillation counter (PerkinElmer). Rates of carbon fixation into
particulate and dissolved organic matter were calculated according to Knap et al. (1996) using the dissolved inorganic carbon
concentration computed for each mesocosm at the beginning of the daily incubations and multiplied by a factor of 1.05 to
correct for the lower uptake of $^{14}$C compared to $^{12}$C.

**2.4 Statistical analysis**

All statistical analyses were performed using R (nlme package). A general least squares (gls) model approach was used to test
the linear effects of the two treatments (temperature, pCO$_2$), and of their interactions on the measured variables (Paul et al.,
2016; Hussherr et al., 2017). The analysis was conducted independently on two different time periods: Phase I (day 0 to day
4) corresponded to the development of the diatom bloom and extended up to the depletion of nitrate, whereas Phase II (day 5
to day 13) corresponded to the declining phase of the bloom in the absence of detectable nitrate. Averages (or time-integration
in the case of primary production) of the response variables were calculated separately over the two phases and were plotted

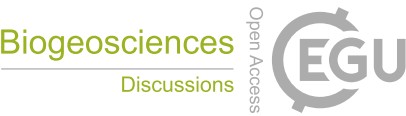

against $pCO_2$. Separate regressions were performed with $pCO_2$ as the continuous factor for each temperature when a
temperature effect or interaction with $pCO_2$ was detected in the gls model. Otherwise, the model included data from both
temperatures and the interaction with $pCO_2$. Normality of the residuals was determined using a Shapiro-Wilk test ($p > 0.05$)
and data were transformed (natural logarithm or square root) if required. As explained by Havenhand et al. (2010), the gradient
approach, instead of treatment replication, is particularly suitable when few experimental units are available such as in large
volume mesocosm experiments.

## 3. Results

### 3.1 Seawater chemistry

Water salinity was $26.52 \pm 0.03$ on day -4 in all mesocosms and remained constant throughout the experiment, averaging
$26.54 \pm 0.02$ on day 13. The TA was practically invariant in the mesocosms, averaging $2057 \pm 2$ µmol $kg_{sw}^{-1}$ on day -4 and
$2058 \pm 2$ µmol $kg_{sw}^{-1}$ on day 13. The pH remained relatively stable throughout the experiment in the pH-controlled treatments,
but decreased slightly during Phase II by an average of $-0.14 \pm 0.07$ units relative to the target $pH_T$ (Fig. 1a). Given a constant
TA, pH variations were accompanied by variations in $pCO_2$, from an average of $1340 \pm 150$ µatm on day -3, and ranging from
564 to 2902 µatm at 10 °C, and from 363 to 2884 µatm at 15 °C on day 0 following the acidification (Fig. 1b). The $pH_T$ in the
Controls (M6 and M11) increased from 7.896 and 7.862 on day 0 at 10 °C and 15 °C, respectively, to 8.307 and 8.554 on day
13, reflecting the balance between $CO_2$ uptake and metabolic $CO_2$ production over the duration of the experiment. On the last
day, $pCO_2$ in all mesocosms ranged from 186 to 3695 µatm at 10 °C, and from 90 to 3480 µatm at 15 °C. The temperature of
the mesocosms in each container remained within $\pm 0.1$ °C of the target temperature throughout the experiment and averaged
$10.04 \pm 0.02$ °C for mesocosms M1 through M6, and $15.0 \pm 0.1$ °C for mesocosms M7 through M12 (Fig. 1c).

### 3.2 Dissolved inorganic nutrient concentrations

Nutrient concentrations averaged $9.1 \pm 0.5$ µmol $L^{-1}$ for $NO_3^-$, $13.4 \pm 0.3$ µmol $L^{-1}$ for $Si(OH)_4$, and $0.91 \pm 0.03$ µmol $L^{-1}$ for
SRP on day 0 (Fig. 1d, e, f). The three nutrients displayed a similar temporal depletion pattern following the development of
the phytoplanktonic bloom. $NO_3^-$ concentrations reached undetectable in all mesocosms on day 5. Likewise, $Si(OH)_4$ fell below
the detection limit between day 1 and 5 in all mesocosms except for those whose $pH_T$ was set at 7.2 and 7.6 at 10 °C (M5 and
M3) and in which $Si(OH)_4$ depletion occurred on day 9. Variations in SRP concentrations followed closely those of $NO_3^-$ in
all mesocosms except again for those set at pH 7.2 and 7.6 in which undetectable values were reached on day 9.

### 3.3 Phytoplankton biomass

Chl *a* concentrations where below 1 µg $L^{-1}$ just after the filling of the mesocosms, and averaged $5.9 \pm 0.6$ µg $L^{-1}$ on day 0 (Fig.
2a). They then quickly increased to reach maximum concentrations around $27 \pm 2$ µg $L^{-1}$ on day $3 \pm 2$, and decreased
progressively until the end of the experiment. During Phase I, results from the gls model show no significant relationships





between the mean Chl *a* concentrations and $pCO_2$ at the two temperatures tested but significantly higher Chl *a* values at 15 °C
than at 10 °C (Fig. 2b; Table 1). During this phase, the accumulation rate of Chl *a* was positively affected by temperature, but
not by the $pCO_2$ gradient (Fig. 3a; Table 2). The maximum Chl *a* concentrations reached during the bloom were not affected
by the two treatments (Fig. 3b; Table 2). During Phase II, we observed no significant effect of increasing $pCO_2$ on the mean
Chl *a* concentrations at the two temperatures tested. Nevertheless, during that phase, the mean Chl *a* concentrations decreased
from $18.2 \pm 0.9$ µg L$^{-1}$ at 10 °C to $12.4 \pm 0.7$ µg L$^{-1}$ at 15 °C, suggesting a faster loss of the pigments following the depletion
of $NO_3^-$.

**3.4 Phytoplankton size-class**

Nanophytoplankton abundance varied from $8 \pm 1 \times 10^6$ cells L$^{-1}$ on day 0 to an average maximum of $36 \pm 10 \times 10^6$ cells L$^{-1}$ at
the peak of the bloom (Fig. 2d). At both temperatures, nanophytoplankton abundance increased until at least days 2 or 4 and
decreased or remained stable thereafter. The strong correlation between the nanophytoplankton abundance and Chl *a* ($r^2 = 0.82$,
$p < 0.001$) suggests that this phytoplankton size class was responsible for most of the biomass build-up in the mesocosms. As
observed for the mean Chl *a* concentration, the mean abundance of nanophytoplankton was not significantly affected by the
$pCO_2$ gradient at the two temperatures investigated during Phase I, but showed higher values at 15 °C ($31 \pm 3 \times 10^6$ cells L$^{-1}$)
than at 10 °C ($13 \pm 2 \times 10^6$ cells L$^{-1}$) (Fig. 2e; Table 1). Likewise, the growth rate of nanophytoplankton during Phase I was
not influenced by the $pCO_2$ gradient at the two temperatures but was significantly higher in the warm treatment (Fig. 3c; Table
2). During Phase II, no relationship was found between the mean nanophytoplankton abundance and the $pCO_2$ gradient at the
two temperatures and no temperature effect was observed (Fig. 2f; Table 3).
Initial abundance of photosynthetic picoeukaryotes was $10 \pm 2 \times 10^6$ cells L$^{-1}$, accounting for more than 80 % of total plankton
cells in the 0.2–20 µm size fraction. The abundance of this plankton size fraction decreased slightly through Phase I and their
number remained relatively stable at $3.3 \pm 0.2 \times 10^6$ cells L$^{-1}$ throughout Phase II (Fig. 2g). We found no relationship between
the abundance of picoeukaryotes and the $pCO_2$ gradient at the two temperatures investigated during both Phases I and II, and
no temperature effect was observed either (Fig. 2h, i; Tables 1 and 3).
Picocyanobacteria exhibited a different pattern than the nanophytoplankton and picoeukaryotes (Fig. 2j). Their abundance was
initially low ($1.7 \pm 0.3 \times 10^6$ cells L$^{-1}$ on day 0), remained relatively stable during Phase I, and increased rapidly during Phase
II, accounting for ca. 50 % of the total picophytoplankton cell counts toward the end of the experiment. During Phase I, the
mean picocyanobacteria abundance was not influenced by the $pCO_2$ gradient but was higher at 15 °C ($1.4 \pm 0.2 \times 10^6$ cells L$^{-1}$)
than at 10 °C ($0.95 \pm 0.05 \times 10^6$ cells L$^{-1}$) (Fig. 2k; Table 1). During Phase II, their mean abundance remained higher at 15 °C
($4.5 \pm 0.3 \times 10^6$ cells L$^{-1}$) than at 10 °C ($2.6 \pm 0.1 \times 10^6$ cells L$^{-1}$), and again no significant effect of $pCO_2$ was detected (Fig
2l; Table 3).



**3.5 Phytoplankton taxonomy**

The taxonomic composition of the planktonic assemblage larger than 2 µm was identical in all treatments at the beginning of the experiment, and was mainly composed of the cosmopolitan chain-forming centric diatom *Skeletonema costatum* (*S. costatum*) and the cryptophyte *Plagioselmis prolonga* var. *nordica* (Fig. 4). At the peak of the blooms (maximum Chl *a* concentrations), the species composition did not vary between the $pCO_2$ treatments and between the two temperatures tested. *S. costatum* was the dominant species in all mesocosms (70–90 % of the total number of eukaryotic cells), except for one mesocosm (M3, pH 7.6 at 10 °C) where a mixed dominance of *Chrysochromulina* spp. (a prymnesiophyte of 2–5 µm) and *S. costatum* was observed (Fig. 4a). *S. costatum* accounted for 80–90 % of the total eukaryotic cell counts in all mesocosms at the end of the experiment carried out at 10 °C. At 15 °C, the composition of the assemblage had shifted toward a dominance of unidentified flagellates and choanoflagellates (2–20 µm) in all mesocosms with these two groups accounting for 55–80 % of the total cell counts while diatoms showed signs of loss of viability as indicated by the presence of empty frustules (Fig. 4b).

**3.6 Primary production**

$P_P$ increased in all mesocosms during Phase I of the experiment, in parallel with the increase in Chl *a* (Fig. 5a). $P_P$ maxima were attained on days 3–4, except for the 15 °C Control (M11) where $P_P$ peaked on day 1. We found no significant effect of the $pCO_2$ gradient on the time-integrated $P_p$ at the two temperatures during both Phases I and II (Fig. 5b, c; Tables 1 and 3), but time-integrated $P_P$ was higher at 15 °C than at 10 °C during Phase I and lower at 15 °C than at 10 °C during Phase II (Tables 1 and 3). Similar opposite responses to warming were observed when normalizing $P_P$ per unit of Chl *a* (Fig. 5g, h, i). Initial Chl *a*-normalized $P_P$ values were $3.3 \pm 0.5$ µmol C (µg Chl *a*)$^{-1}$ d$^{-1}$ and reached maxima between $3.7 \pm 0.3$ µmol C (µg Chl *a*)$^{-1}$ d$^{-1}$ and $6.0 \pm 0.7$ µmol C (µg Chl *a*)$^{-1}$ d$^{-1}$ at 10 °C and 15 °C, respectively. These values then decreased to $2.2 \pm 0.6$ µmol C (µg Chl *a*)$^{-1}$ d$^{-1}$ and $0.9 \pm 0.2$ µmol C (µg Chl *a*)$^{-1}$ d$^{-1}$ on the last day of the experiment (Fig. 5g). During Phase I, the mean Chl *a*-normalized $P_P$ was significantly higher under warming, but as observed for Chl *a* concentrations and $P_P$, was not affected by the $pCO_2$ gradient (Fig. 5h; Table 1). During Phase II, the mean Chl *a*-normalized $P_P$ decreased with increasing temperature, with values of $2.2 \pm 0.2$ µmol C (µg Chl *a*)$^{-1}$ d$^{-1}$ and $1.6 \pm 0.1$ µmol C (µg Chl *a*)$^{-1}$ d$^{-1}$ at 10 °C and 15 °C, respectively (Fig. 5i; Table 3). No significant effect of $pCO_2$ was detected.

$P_D$ was low at the beginning of the experiment, averaging $1.5 \pm 0.4$ µmol C L$^{-1}$ d$^{-1}$, increased progressively during Phase I to reach values of up to 48 µmol C L$^{-1}$ d$^{-1}$ on days 4 and 5 at the beginning of Phase II, and decreased thereafter (Fig. 5d). Time-integrated $P_D$ was affected neither by the $pCO_2$ gradient at the two temperatures tested nor by temperature during the two Phases (Fig. 5e, f; Tables 1 and 3). Chl *a*-normalized $P_D$ was low on day 0, averaging $0.3 \pm 0.1$ µmol C (µg Chl *a*)$^{-1}$ d$^{-1}$, reached maximum values of $1.4 \pm 0.5$ µmol C (µg Chl *a*)$^{-1}$ d$^{-1}$ at the beginning of Phase II, and decreased to $0.3 \pm 0.4$ µmol C (µg Chl *a*)$^{-1}$ d$^{-1}$ by the end of the experiment (Fig. 5j). During Phases I and II, the mean Chl *a*-normalized $P_D$



were affected neither by the temperature and $pCO_2$ gradient, nor by the interaction between those factors (Fig. 5k, l; Tables 1
and 3).
Figure 6 shows the influence of the treatments on maximum $P_P$ and $P_D$ as well as on the time-integrated $P_P$ and $P_D$ over the full
length of the experiment. We found no effect of the $pCO_2$ gradient on the maximum $P_P$ values at the two temperatures tested,
but warming increased the maximum $P_P$ values from $66 \pm 13$ µmol C $L^{-1}$ $d^{-1}$ to $126 \pm 8$ µmol C $L^{-1}$ $d^{-1}$ (Fig. 6a; Table 4). The
time-integrated $P_P$ over the full duration of the experiment was not affected by the $pCO_2$ gradient or the increase in temperature
(Fig. 6b; Table 4). The absence of temperature effect results from the countervailing responses in time-integrated $P_P$ between
Phases I and II. The maximum $P_D$ values were significantly affected by the treatments (Fig 6c; Table 4). Maximum $P_D$
decreased with increasing $pCO_2$ at in situ temperature but warming cancelled this effect (antagonistic effect). Nevertheless,
the time-integrated $P_D$ over the whole experiment did not vary significantly between treatments, although a decreasing
tendency with increasing $pCO_2$ at 10 °C and an increasing tendency with warming can be seen in Fig. 6d (Table 4).
**4. Discussion**
**4.1 General characteristics of the bloom**
A phytoplankton bloom, numerically dominated by the centric diatom *S. costatum*, took place in all mesocosms, regardless of
treatments (Fig. 4). *S. costatum* is a common phytoplankton species in the St. Lawrence Estuary and in coastal waters (Kim et
al., 2004; Starr et al., 2004; Annane et al., 2015). The length of the experiment (13 days) allowed us to capture both the
development and declining phases of the bloom. The exponential growth phases lasted 1–4 days depending on the treatments,
but maximal Chl *a* concentrations were reached only after 7 days in two of the twelve mesocosms (Fig. 2a). The suite of
measurements and statistical tests conducted did not provide any clues as to the underlying causes for the lower rates of biomass
accumulation measured in these two mesocosms. Since statistical analyses conducted with or without these two apparent
outliers gave similar results, they were not excluded from the analyses.
During the development phase of the bloom, the concentration of all three monitored nutrients decreased, with $NO_3^-$ and
$Si(OH)_4$ reaching undetectable values. This nutrient co-depletion is consistent with results from previous studies suggesting a
co-limitation of diatom blooms by these two nutrients in the St. Lawrence Estuary (Levasseur et al., 1987, 1990). Variations
in $P_P$ roughly followed changes in Chl *a*, and, as expected, the maximum Chl *a*-normalized $P_P$ ($5 \pm 2$ µmol C (µg Chl *a*)$^{-1}$ $d^{-1}$)
was reached during the exponential growth phase in all mesocosms. Decreases in total phytoplankton abundances and $P_P$
followed the bloom peaks and the timing of the $NO_3^-$ and $Si(OH)_4$ depletions. A clear succession in phytoplankton size classes
characterized the experiment. Nanophytoplankton cells were initially present in low abundance and became more numerous
as the *S. costatum* diatom bloom developed. The strong correlation ($r^2 = 0.83$, $p < 0.001$) between the abundance of
nanophytoplankton and *S. costatum* enumeration suggests that this cell size class can be used as a proxy of *S. costatum* counts
in all mesocosms throughout the experiment. Nanophytoplankton cells accounted for $79 \pm 7$ % of total counts of cells < 20 µm
on the day of the maximum Chl *a* concentration. Accordingly, nanophytoplankton exhibited the same temporal trend as Chl *a*



concentrations. During Phase II, nanophytoplankton abundances remained roughly stable at in situ temperature but decreased
at 15 °C. Photosynthetic picoeukaryotes were originally abundant and decreased throughout the experiment whereas
picocyanobacteria abundances increased during Phase II. This is a typical phytoplankton succession pattern for temperate
systems where an initial diatom bloom growing essentially on allochthonous nitrate gives way to smaller species growing on
regenerated forms of nitrogen (Taylor et al., 1993).
**4.2 Phase I (Diatom bloom development)**
Our results show no significant effect of increasing $pCO_2$/decreasing pH on the mean abundance and net accumulation rate of
the diatom-dominated nanophytoplankton assemblage during the development of the bloom (Figs. 2e and 3c). These results
suggest that *S. costatum*, the species accounting for most of the biomass accumulation during the bloom, neither benefited
from the higher $pCO_2$ nor was negatively impacted by the lowering of pH. Assuming that *S. costatum* was also responsible for
most of the carbon fixation during the bloom development phase, the absence of effect on $P_P$ and Chl *a*-normalized $P_P$ following
increases in $pCO_2$ brings additional support to our conclusion. *S. costatum* operates a highly efficient CCM, minimizing the
potential benefits of thriving in high $CO_2$ waters (Trimborn et al., 2009). This may explain why the strain present in the LSLE
did not benefit from the higher $pCO_2$ conditions. Likewise, a mesocosm experiment conducted in the coastal North Sea showed
no significant effect of increasing $pCO_2$ on carbon fixation during the development of the spring diatom bloom (Bach et al.,
2017; Eberlein et al., 2017).
In addition to the aforementioned insensitivity to increasing $pCO_2$, our results point towards a strong resistance of *S. costatum*
to severe pH decline. During our study, surprisingly constant rates of Chl *a* accumulation and nanophytoplankton growth (Fig.
3a, c), as well as maximum $P_P$ (Fig. 6a), were measured during the development phase of the bloom over a range of $pH_T$
extending from 8.6 to 7.2 (Fig. 1a). In a recent effort to estimate the causes and amplitudes of short-term variations in $pH_T$ in
the LSLE, Mucci et al. (2017) showed that $pH_T$ in surface waters was constrained within a range of 7.85 to 7.93 during a 50-
h survey over two tidal cycles at the head of the Laurentian Channel. It is notable that even the upwelling of water from 100 m
depth or of low-oxygen LSLE bottom water would not decrease $pH_T$ beyond ca. 7.75 and 7.62, respectively (Mucci et al., 2017
and references therein). Our results show that the phytoplankton assemblage responsible for the fall bloom may tolerate even
greater $pH_T$ excursions. In the LSLE, such conditions may arise when the contribution of the low $pH_T$ (7.12) freshwaters of
the Saguenay River to the LSLE surface waters is amplified during the spring freshet. However, considering that comparable
studies conducted in different environments have reported negative effects of decreasing pH on diatom biomass accumulation
(Hare et al., 2007; Hopkins et al., 2010; Schulz et al., 2013), it cannot be concluded that all diatom species thriving in the
LSLE are insensitive to acidification.
In contrast to the $pCO_2$ treatment, warming affected the development of the bloom in several ways. Increasing temperature by
5 °C significantly increased the accumulation rate of Chl *a*, the nanophytoplankton growth rate, as well as the time-integrated
$P_P$ and Chl *a*-normalized $P_P$ during Phase I of the bloom. The positive effects of warming on the Chl *a*-normalized $P_P$ during
the development phase of the bloom most likely reflect the sensitivity of photosynthesis to temperature (Sommer and





Lengfellner, 2008; Kim et al., 2013). It could also be related to optimal growth temperatures, which are often higher than in
situ temperatures in marine phytoplankton (Thomas et al., 2012; Boyd et al., 2013). In support of this hypothesis, previous
studies have reported optimal growth temperatures of 20–25 °C for *S. costatum*, which is 5–10 °C higher than the warmer
treatment investigated in our study (Suzuki and Takahashi, 1995; Montagnes and Franklin, 2001). Extrapolating results from
a mesocosm experiment to the field is not straightforward, as little is known of the projected warming of the upper waters of
the LSLE in the next decades. In the Gulf of St. Lawrence, positive temperature anomalies in surface waters have varied from
0.25 to 0.75 °C per decade between 1985 and 2013 (Larouche and Galbraith, 2016). In the LSLE, warming of surface waters
will likely result from a complex interplay between heat transfer at the air-water interface and variations in vertical mixing and
upwelling of the cold intermediate layer at the head of the Estuary (Galbraith et al., 2014). Considering current uncertainties
regarding future warming of the LSLE, studies should be conducted over a wider range of temperatures in order to better
constrain the potential effect of warming on the development of the blooms in the LSLE.
Picoeukaryotes showed a more or less gradual decrease in abundance during Phase I, and our results show that this decline
was not influenced by the increases in $pCO_2$ (Fig. 2g; Table 1). Picoeukaryotes are expected to benefit from high $pCO_2$
conditions even more so than diatoms as $CO_2$ can passively diffuse through their relatively thin boundary layer precluding the
necessity of a costly uptake mechanism such as a CCM (Schulz et al., 2013). This hypothesis has been supported by several
studies showing a stimulating effect of $pCO_2$ on picoeukaryote growth (Bach et al., 2016; Hama et al., 2016; Schulz et al.,
2017 and references therein). On the other hand, in nature, the abundance of picoeukaryotes generally results from a delicate
balance between cell division rates and cell losses through microzooplankton grazing and viral attacks. The few experiments,
including the current study, reporting the absence or a modest effect of increasing $pCO_2$ on the abundance of eukaryotic
picoplankton attribute their observations to an increase in nano- and microzooplankton grazing (Rose et al., 2009; Neale et al.,
2014). During our experiment, the biomass of microzooplankton increased with increasing $pCO_2$ by ca. 200–300 % at the two
temperatures tested (Ferreyra and Lemli, unpubl. data). Thus, it is possible that a positive effect of increasing $pCO_2$ and
warming on picoeukaryote abundances might have been masked by higher picoeukaryote losses due to increased
microzooplankton grazing.

### 370 4.3 Phase II (declining phase of the bloom)

The gradual decrease in nanophytoplankton abundances coincided with an increase in the abundance of picocyanobacteria
(Fig. 2j). Regardless of temperature, the picocyanobacteria abundance during Phase II was unaffected by the increase in $pCO_2$
over the full range investigated (Fig. 2l; Table 3). The lack of positive response of picocyanobacteria to elevated $pCO_2$ was
somewhat surprising considering that they have less efficient CCMs than diatoms (Schultz et al., 2013). Accordingly, several
studies have reported a stimulation of the net growth rate of picocyanobacteria under elevated $pCO_2$ in different environments
(coastal Japan, Mediterranean Sea, and Raunejforden in Norway) and under different nutrient regimes, i.e. bloom and post-
bloom conditions (Hama et al., 2016; Sala et al., 2016; Schulz et al., 2017). Consistent with our observations, however, Law
et al. (2012) and Lomas et al. (2012) observed no direct effect of elevated $pCO_2$ on the net growth of picocyanobacteria during

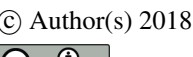



studies conducted in the Subtropical North Atlantic and the South Pacific. In our study, a potential increase in grazing pressure,
following the rise in heterotrophic nanoflagellates abundance (e.g. choanoflagellates; Fig. 4b) measured under high $pCO_2$ and
warmer conditions, could explain the ostensible absence of stimulation of picocyanobacteria by increasing $pCO_2$. Despite the
absence of grazing measurements during our study, our results support the hypothesis that the potential for increased
picocyanobacteria population growth under elevated $pCO_2$ and temperature is partially dependent on different grazing
pressures (Fu et al., 2007).
Warming, not acidification, affected carbon fixation during the declining phase of the bloom. In our study, the time-integrated
primary production and Chl $a$-normalized primary production were not significantly affected by the increase in $pCO_2$ during
Phase II at the two temperatures tested (Fig. 5, Table 3). This result is surprising since nitrogen-limited cells have been shown
to be more sensitive to acidification, resulting in a reduction in carbon fixation rates due to higher respiration (Wu et al., 2010;
Gao and Campbell, 2014; Raven et al., 2014). Although our measurements do not allow to discriminate between the
contributions of the different phytoplankton size classes to carbon fixation, we can speculate that diatoms, which were still
abundant during Phase II, contributed to a significant fraction of the primary production. If so, these results suggest that *S.*
*costatum* remained insensitive to OA even under nutrient stress. However, in contrast to Phase I, increasing the temperature
by 5 °C during Phase II significantly reduced $P_P$ and the Chl $a$-normalized $P_P$. Dark phytoplankton respiration rates generally
increase with temperature (Butrón et al., 2009; Robarts and Zohary, 1987). The warming-induced decrease in carbon fixation
measured during Phase II may thus result from an increase in respiration by the nitrogen-limited diatoms.

### 4.4 Effect of the treatments on primary production over the full experiment

As mentioned above, increasing $pCO_2$ had no effect on time-integrated $P_P$ during the two phases of the bloom, but warming
resulted respectively in a positive and negative effect during Phases I and II. As a result, primary production rates integrated
over the whole duration of the experiment were not significantly different between the two temperatures tested. Although not
statistically significant, the time-integrated $P_{DS}$ over the full experiment display a slight decrease with increasing $pCO_2$ at 10 °C
and overall higher values in the warmer treatment (Fig. 6d; Table 4). Previous studies have reported increases of dissolved
organic carbon (DOC) exudation (Engel et al., 2013), but also decreasing DOC concentrations at elevated $pCO_2$ under nitrate
limitation (Yoshimura et al., 2014). The increase in DOC exudation is attributed to a stimulation of photosynthesis resulting
from its sensitivity to higher $pCO_2$ (Engel et al. 2013), but the causes for a decrease in DOC concentrations at high $pCO_2$ are
less clear and potentially attributable to an increase in transparent exopolymer particle (TEP) production (Yoshimura et al,
2014). Elevated TEP production under high $pCO_2$ conditions has been measured both at the peak of a bloom in a mesocosm
study (Engel et al., 2014), and in post-bloom nutrient depleted conditions (MacGilchrist et al., 2014). However, during our
study, TEP production decreased under high $pCO_2$ (Gaaloul, 2017). Thus, the apparent decrease in $P_D$ cannot be attributed to
a greater conversion of exuded dissolved carbohydrate into TEP. The apparent rise in $P_D$ under warming is consistent with
previous studies reporting similar increases in phytoplankton dissolved carbon release with temperature (Morán et al., 2006;
Engel et al., 2011). Although these apparent changes in $P_D$ with increasing $pCO_2$ and warming require further investigations,



they suggest that a larger proportion (ca. 15 % of $P_T$ at 15 °C compared to 10 % at 10 °C) of the newly fixed carbon could be
exuded and become available for heterotrophic organisms under warmer conditions.

## 5. Conclusion

Our results reveal a remarkable resistance of the different phytoplankton size classes to the large range of $pCO_2$/pH investigated
during our study. It is noteworthy that the plankton assemblage was submitted to decreases in pH far exceeding those that they
are regularly exposed to in the LSLE. The resistance of *S. costatum* to the $pCO_2$ treatments suggests that the acidification of
surface waters of the LSLE will not affect the development rate and the amplitude of fall blooms dominated by this species.
Photosynthetic picoeukaryotes and picocyanobacteria thriving alongside the blooming diatoms were also insensitive to
acidification. In contrast to the $pCO_2$ treatments, warming the water by 5 °C had multiple impacts on the development and
decline of the bloom. The 5 °C warming hastened the development of the diatom bloom (albeit with no increase in total cells
number) and increased the abundance of picocyanobacteria. These temperature-induced variations in the phytoplankton
assemblage were accompanied, respectively, by higher then lower $P_P$ during the development and declining phases of the
diatom bloom. Due to these contrasting responses, warming had no net effect on $P_P$ over the full temporal scale of the
experiment. Overall, our results indicate that warming could have more important impacts than acidification on phytoplankton
bloom development in the LSLE in the next decades. Future studies should be conducted and specifically designed to better
constrain the potential effects of warming on phytoplankton succession and primary production in the LSLE.
*Data availability.* The data have been submitted to be freely accessible via https://issues.pangaea.de/browse/PDI-16607, or
can be obtained by contacting the author (robin.benard.1@ulaval.ca).
*Author contributions.* R. Bénard was responsible for the experimental design elaboration, data sampling and processing, and
the redaction of this article. Several co-authors supplied specific data included in this article, and all co-authors contributed to
this final version of the article.
*Competing interests.* The authors declare that they have no conflict of interest.

## Acknowledgements

The authors wish to thank the Station Aquicole ISMER, especially Nathalie Morin and her staff, for their support during the
project. We also wish to acknowledge Gilles Desmeules, Bruno Cayouette, Sylvain Blondeau, Claire Lix, Rachel Hussherr,
Liliane St-Amand, Marjolaine Blais, Armelle Simo and Sonia Michaud for their help in setting up, sampling and processing
samples during the experiment. The authors want to thank Jean-Pierre Gattuso for his constructive comments on an earlier
draft of this manuscript. This study was funded by a Team grant from the Fonds de la Recherche du Québec – Nature et
Technologies (FRQNT-Équipe-165335), the Canada Foundation for Innovation, and the Canada Research Chair on Ocean
Biogeochemistry and Climate. This is a contribution to the research programme of Québec-Océan.



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




**Figure 1. Temporal variations over the course of the experiment for: (a) pH$_T$, (b) pCO$_2$, (c) temperature, (d) nitrate, (e) silicic acid,**
**(f) soluble reactive phosphate. For symbol attribution to treatments, see legend.**




**Figure 2. Temporal variations, and averages ± SE during Phase I (day 0 to day 4) and Phase II (day 5 to day 13) for: (a-c)
chlorophyll *a*, (d-f) nanophytoplankton, (g-i) picoeukaryotes, (j-l) picocyanobacteria. For symbol attribution to treatments, see
legend.**





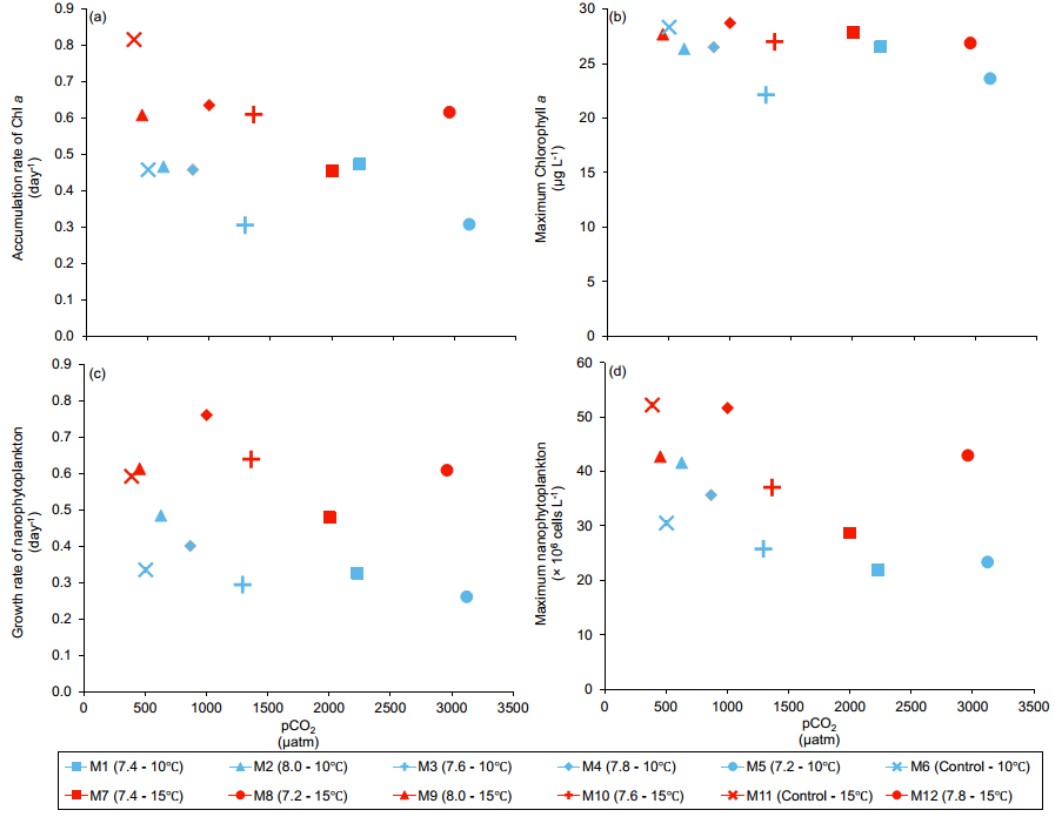

**Figure 3. (a) Accumulation rate of Chl *a* (day 0 to maximum Chl *a* concentration), (b) maximum Chl *a* concentrations, (c) growth rate of nanophytoplankton (day 0 to maximum nanophytoplankton abundance), and (d) maximum nanophytoplankton abundance during the experiment. For symbol attribution to treatments, see legends.**



Figure 4. Relative abundance of 10 groups of protists at the beginning of the experiment (day -4), on the day of maximum Chl *a* concentrations in each mesocosm, and at the end of the experiment (day 13) for (a) 10 °C and (b) 15 °C mesocosms. The group « others » include dinoflagellates, Chlorophyceae, Dictyochophyceae, Euglenophyceae, heterotrophic groups, and unidentified cells. Each bar plot represents a mesocosm at a given time. The bar plot on day -4 represents the initial community assemblage before temperature manipulation and acidification, and is therefore the same for each temperature treatment. For symbol attribution to treatments, see legend.







720 **Figure 5.** Temporal variations, time-integrated or averaged ± SE during Phase I (day 0 to day 4) and Phase II (day 5 to day 13) for:
721 **(a-c) particulate primary production, (d-f) dissolved primary production, (g-i) Chl *a*-normalized particulate primary production, (j-**
722 **l) Chl *a*-normalized dissolved primary production. For symbol attribution to treatments, see legend.**



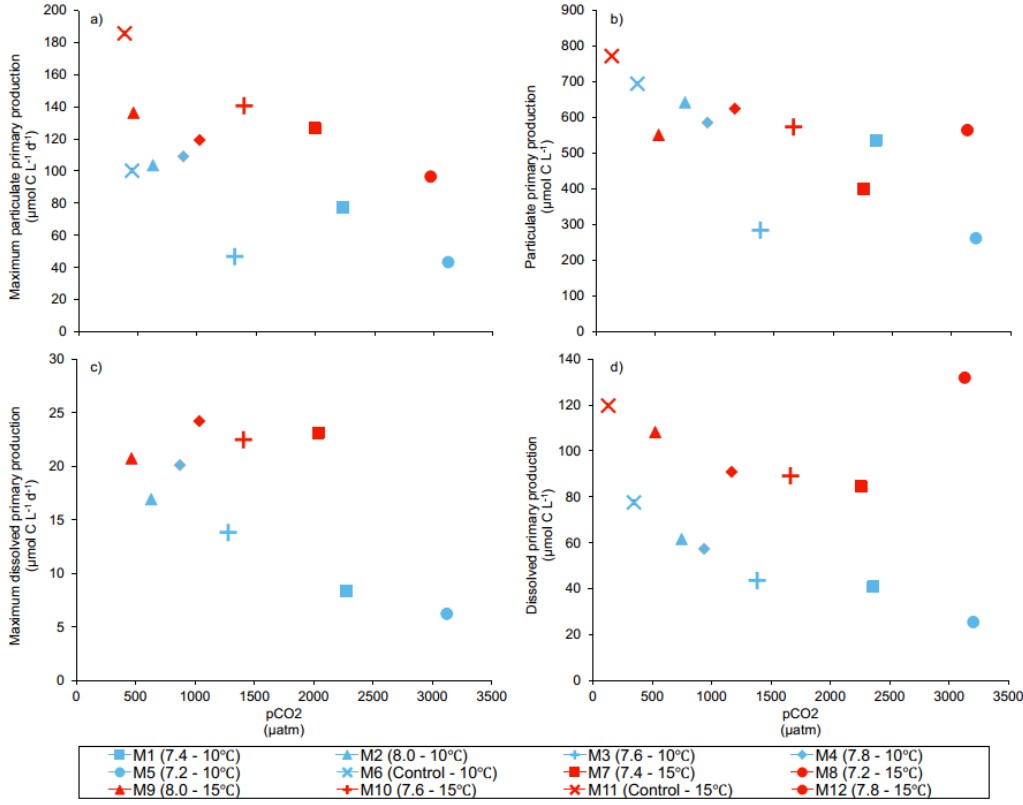

**Figure 6. (a) Maximum particulate primary production, (b) time-integrated particulate primary production (c) maximum dissolved primary production, and (d) time-integrated dissolved primary production over the full course of the experiment (day 0 to day 13). For symbol attribution to treatments, see legend.**





**Table 1. Results of the generalized least squares models (gls) tests for the effects of temperature. pCO₂. and their interaction during Phase I (day 0 to day 4). Separate analysis with pCO₂ as a continuous factor were performed when temperature had a significant effect. Chl *a* concentration, nanophytoplankton abundance, picoeukaryote abundance, picocyanobacteria abundance, particulate and dissolved primary production, and Chl *a*-normalized particulate and dissolved primary production. Significant results are in bold. \*p < 0.05.**

| Response Variable | Factor | df | t-value | p-value |
|---|---|---|---|---|
| Mean Chl *a* concentration ($\mu$g L$^{-1}$) | Temperature | **8** | **2.307** | **0.050\*** |
| | pCO₂ (10 °C) | 4 | -1.362 | 0.245 |
| | pCO₂ (15 °C) | 4 | -1.263 | 0.275 |
| Mean nanophytoplankton abundance ($\times$ 10$^6$ cells L$^{-1}$) | Temperature | **8** | **2.980** | **0.018\*** |
| | pCO₂ (10 °C) | 4 | -2.729 | 0.053 |
| | pCO₂ (15 °C) | 4 | -1.231 | 0.286 |
| Mean picoeukaryote abundance ($\times$ 10$^6$ cells L$^{-1}$) | Temperature | 8 | 1.157 | 0.281 |
| | pCO₂ | 8 | -1.070 | 0.316 |
| | pCO₂ $\times$ Temperature | 8 | 1.085 | 0.309 |
| Mean picocyanobacteria abundance ($\times$ 10$^6$ cells L$^{-1}$) | Temperature | **8** | **3.066** | **0.015\*** |
| | pCO₂ (10 °C) | 4 | 0.125 | 0.907 |
| | pCO₂ (15 °C) | 4 | -2.268 | 0.086 |
| Particulate primary production ($\mu$mol C L$^{-1}$) | Temperature | **8** | **2.690** | **0.028\*** |
| | pCO₂ (10 °C) | 4 | -1.617 | 0.181 |
| | pCO₂ (15 °C) | 4 | -0.992 | 0.378 |
| Dissolved primary production ($\mu$mol C L$^{-1}$) | Temperature | 8 | 0.756 | 0.472 |
| | pCO₂ | 8 | -0.901 | 0.394 |
| | pCO₂ $\times$ Temperature | 8 | 0.956 | 0.367 |
| Chl *a*-normalized particulate primary production ($\mu$mol C ($\mu$g Chl *a*)$^{-1}$ d$^{-1}$) | Temperature | **8** | **2.592** | **0.032\*** |
| | pCO₂ (10 °C) | 4 | -1.467 | 0.216 |
| | pCO₂ (15 °C) | 4 | -0.840 | 0.448 |



| Chl $a$-normalized dissolved primary production ($\mu$mol C ($\mu$g Chl $a$)$^{-1}$ d$^{-1}$) | Temperature | 8 | -0.350 | 0.735 |
|---|---|---|---|---|
| | pCO$_2$ | 8 | -0.397 | 0.702 |
| | pCO$_2$ $\times$ Temperature | 8 | 0.522 | 0.616 |

733





**Table 2. Results of the generalized least squares models (gls) tests for the effects of temperature, pCO₂ and their interaction. Separate analysis with pCO₂ as a continuous factor were performed when temperature had a significant effect. Accumulation rate of Chl *a* (day 0 to maximum Chl *a* concentration), maximum Chl *a* concentration, growth rate of nanophytoplankton (day 0 to maximum nanophytoplankton abundance), and maximum nanophytoplankton abundance. Significant results are in bold. *p < 0.05.**

| Response Variable | Factor | df | t-value | p-value |
|---|---|---|---|---|
| Accumulation rate of Chl *a* (day$^{-1}$) | Temperature | **8** | **2.679** | **0.028*** |
| | pCO$_2$ (10 °C) | 4 | -1.476 | 0.214 |
| | pCO$_2$ (15 °C) | 4 | -1.759 | 0.154 |
| Maximum Chl *a* concentration (μg L$^{-1}$) | Temperature | 8 | 1.305 | 0.228 |
| | pCO$_2$ | 8 | -0.387 | 0.709 |
| | pCO$_2$ × Temperature | 8 | 0.022 | 0.983 |
| Growth rate of nanophytoplankton (day$^{-1}$) | Temperature | **8** | **2.534** | **0.035*** |
| | pCO$_2$ (10 °C) | 4 | -0.882 | 0.403 |
| | pCO$_2$ (15 °C) | 4 | 0.601 | 0.564 |
| Maximum nanophytoplankton abundance (× 10$^6$ cells L$^{-1}$) | Temperature | 8 | 1.380 | 0.205 |
| | pCO$_2$ | 8 | -0.735 | 0.484 |
| | pCO$_2$ × Temperature | 8 | 0.302 | 0.770 |






**Table 3. Results of the generalized least squares models (gls) tests for the effects of temperature, $pCO_2$, and their interaction during**
**Phase II (day 5 to day 13). Separate analysis with $pCO_2$ as a continuous factor were performed when temperature had a significant**
**effect. Chl *a* concentration, nanophytoplankton abundance, picoeukaryote abundance, picocyanobacteria abundance, particulate**
**and dissolved primary production, and Chl *a*-normalized particulate and dissolved primary production. Significant results are in**
**bold. \*p < 0.05, \*\*p < 0.01, \*\*\*p < 0.001.**

| Response Variable | Factor | df | t-value | p-value |
|---|---|---|---|---|
| Mean Chl *a* concentration ($\mu$g L$^{-1}$) | Temperature | 8 | -3.600 | **0.007\*\*** |
|  | $pCO_2$ (10 °C) | 4 | -2.724 | 0.073 |
|  | $pCO_2$ (15 °C) | 4 | -1.263 | 0.275 |
| Mean nanophytoplankton abundance ($\times$ 10$^6$ cells L$^{-1}$) | Temperature | 8 | -1.465 | 0.181 |
|  | $pCO_2$ | 8 | -1.539 | 0.162 |
|  | $pCO_2 \times$ Temperature | 8 | 1.003 | 0.345 |
| Mean picoeukaryotes abundance ($\times$ 10$^6$ cells L$^{-1}$) | Temperature | 8 | 0.581 | 0.577 |
|  | $pCO_2$ | 8 | 0.294 | 0.776 |
|  | $pCO_2 \times$Temperature | 8 | -0.698 | 0.505 |
| Mean picocyanobacteria abundance ($\times$ 10$^6$ cells L$^{-1}$) | Temperature | 8 | 6.107 | **<0.001\*\*\*** |
|  | $pCO_2$ (10 °C) | 4 | 0.401 | 0.709 |
|  | $pCO_2$ (15 °C) | 4 | -2.347 | 0.079 |
| Particulate primary production ($\mu$mol C L$^{-1}$) | Temperature | 8 | **-2.248** | **0.012\*** |
|  | $pCO_2$ (10 °C) | 4 | -2.186 | 0.094 |
|  | $pCO_2$ (15 °C) | 4 | -2.390 | 0.075 |
| Dissolved primary production ($\mu$mol C L$^{-1}$) | Temperature | 8 | 1.154 | 0.282 |
|  | $pCO_2$ | 8 | -1.701 | 0.127 |
|  | $pCO_2 \times$ Temperature | 8 | 1.369 | 0.208 |
| Chl *a*-normalized particulate primary production ($\mu$mol C ($\mu$g Chl *a*)$^{-1}$ d$^{-1}$) | Temperature | 8 | -3.387 | **0.010\*\*** |
|  | $pCO_2$ (10 °C) | 4 | -2.226 | 0.090 |
|  | $pCO_2$ (15 °C) | 4 | -0.366 | 0.733 |



| Chl *a*-normalized dissolved primary production (μmol C (μg Chl *a*)$^{-1}$ d$^{-1}$) | Temperature | 8 | 1.973 | 0.073 |
|---|---|---|---|---|
| | pCO$_2$ | 8 | -1.838 | 0.103 |
| | pCO$_2$ × Temperature | 8 | 1.860 | 0.100 |






**Table 4. Results of the generalized least squares models (gls) tests for the effects of temperature, pCO₂ and their interaction. Separate analysis with pCO₂ as a continuous factor were performed when temperature had a significant effect. Maximum particulate and dissolved primary production, and time-integration over the full duration of the experiment (day 0 to day 13). Natural logarithm transformation is indicated in parentheses when necessary, significant results are in bold. \*p < 0.05, \*\*p < 0.01.**

| Response Variable | Factor | Df | t-value | p-value |
|---|---|---|---|---|
| Maximum particulate primary production ($\mu$mol C L$^{-1}$ d$^{-1}$) | Temperature | 8 | **2.466** | **0.039\*** |
| | pCO$_2$ (10 °C) | 4 | -2.328 | 0.080 |
| | pCO$_2$ (15 °C) | 4 | -2.394 | 0.075 |
| Time-integrated particulate primary production ($\mu$mol C L$^{-1}$ d$^{-1}$) | Temperature | 8 | -0.055 | 0.958 |
| | pCO$_2$ (10 °C) | 4 | -1.300 | 0.230 |
| | pCO$_2$ (15 °C) | 4 | 0.801 | 0.446 |
| (Log) Maximum dissolved primary production ($\mu$mol C L$^{-1}$) | Temperature | 8 | -0.659 | 0.528 |
| | pCO$_2$ | 8 | **-3.342** | **0.010\*\*** |
| | pCO$_2$ × Temperature | 8 | **2.858** | **0.021\*** |
| Time-integrated dissolved primary production ($\mu$mol C L$^{-1}$) | Temperature | 8 | 1.687 | 0.130 |
| | pCO$_2$ | 8 | -2.153 | 0.063 |
| | pCO$_2$ × Temperature | 8 | 1.880 | 0.097 |

749