# Peer review of "Experimental assessment of the sensitivity of an estuarine phytoplankton fall bloom to acidification and warming"

_Biogeosciences, 2018_

## Referee Comment (RC1) · Anonymous Referee #1 · 25 Mar 2018

General comments:

The manuscript by Benard et al. describes the results from a mesocosm experiment that was designed to investigate the responses of a natural phytoplankton community to warming and acidification. The authors observed a clear stimulation of phytoplankton growth by temperature whereas acidification had no or only a minor effect. Although many experimental studies have been conducted in recent years to investigate phytoplankton responses to warming or to acidification little is known about combined effects. The data provided by this study are thus potentially valuable and interesting. However, important information is lacking in the current version and need to be included and

discussed to improve the value of this manuscript.

The set-up of the experiment was designed to keep pH in the acidified mesocosms constant, yielding a decrease of pH after the bloom. This differs to the natural environment where a phytoplankton bloom can substantially modify (increase) pH. It also differs to the earlier mesocosms experiments that the authors reference in their discussion. I suggest that the authors discuss implications of the differences in the set-up of experiments.

Nutrient concentration and irradiance are main factors controlling phytoplankton growth in seawater. The authors should asses how these factors may have affected cell growth and primary production. This includes:

1. Add a drawing of the set-up and placement of mesocosms and treatments within the container. Containers often bear the risk of self-shading, which would need to be considered.

2. Please give absolute values for irradiance instead of %shading. What was the light:dark cycle? Since primary production measurements were carried out over 24h, some incubation hours will have been in the dark if natural sunlight was applied. It is important to inform about the potential role of dark respiration during the incubations. Since light intensity has been shown to co-affect phytoplankton responses to acidification and warming, it is absolutely necessary to show and discuss the light climate in mesocosms and incubations.

3. It is important to consider that net primary production was measured. Hence, responses to warming and acidification may not only be related to photosynthetic production but also to respiration processes. Please discuss.

Given that the authors did not add nutrients to the natural seawater, the strong increase in biomass (from 10 to up to 30 $\mu$g/L Chl a in one day) after incubation is very surprising. What could have limited phytoplankton growth in situ? Please discuss.

[Figure]

There was a strong drop in pH prior to the acidification treatment on day -3. What may have been the reason for this drop?

Specific comments:

Line 87: add diameter and height of mesosoms

Line 99ff: add total duration of the experiment to the description

Line 104: give value for initial pCO2

Line 113ff: Add total amount of volume sampled from the mesocosms each day

Line 166: Was the no replicate incubation? Was the error within treatment assessed?

Line 171: Give irradiance values

Line 202: pCO2 was 1340 ± 150 $\mu$atm on day -3; why was the value so high?

Line 211:' The three nutrients displayed a similar temporal depletion pattern following the development of the phytoplanktonic bloom.' I disagree the nutrients in the warm treatments were clearly reduced much faster.

Line 217: 'Chl a concentrations where below 1 $\mu$g L-1 just after the filling of the meso-cosms, and averaged 5.9 ± 0.6 $\mu$g L-1 on day 0' If Chla increased that much regardless of treatment; light limitation or exclusion of zooplankton probably had a major influence of phytoplankton development and should be considered in more depth.

Line 327: The citation of Bach et al 2017 is not accurate as that study didn't determine carbon fixation

Figure 3: Wasn't the Chl a accumulation (day 0 to Chl a max) not much higher in the warm control?

Figure 5 g, f: same axis labelling different figure. . .please check.

---

## Referee Comment (RC2) · Anonymous Referee #2 · 4 Apr 2018

The manuscript of Bénard and collaborators reports on an experiment that has been conducted using indoor mesocosms (2.6 m3) to test for the effect of ocean acidification and warming on the development of a fall phytoplankton bloom in the Lower St. Lawrence Estuary. The experiment setup comprised 2 sets of 6 mesocosms installed in two temperature-controlled containers, that were filled with seawater sieved onto 250 microns. In one container, the water temperature was raised by 5°C compared to the mesocosms installed in the other container (10 vs.  $15^{\circ}$ C). A gradient approach (no replicates) has been considered for pCO2/pH covering a range of pH from 7.2 to 8.6. The experiment lasted 13 days and covered the development of a bloom and its decline. Major conclusions of this study are that pCO2 has no effect on all measured parameters and processes while increasing temperature led to a faster build-up of chlorophyll and higher particulate primary production rates. Overall, this is a very well written manuscript that deals with an important topic. The introduction is well documented and shows that while this topic is of great importance, a fair amount of studies has already been conducted, including studies using in situ mesocosms in various environments. Although I would like to ultimately recommend this manuscript for publication in BG, I am concerned by 3 major aspects of this work and would like the authors to answer these comments. 1) Realism. The authors clearly mention that the surface mixed-layer pCO2 is strongly modulated by biological productivity, yet they decided to run an experiment during which a bloom is produced and where carbonate chemistry has been maintained as constant. This would be acceptable if well explained and discussed, but the problem is that "control" mesocosms were actually not controlled (consider changing their name...) and pH was left increasing while the bloom was forming to (what I consider to be) very high and potentially unrealistic pH (?) values of 8.6. In situ pH was apparently close to 7.8, these "control" mesocosms appear to me as "perturbed"! Besides this major concern, I have to admit I do not understand how carbonate chemistry was controlled. The authors mention that "acidification" was carried out over day -1. On that day, I actually also observe a sudden increase of pH for the "controls", pH8 and pH7.8... How did that happen? Naturally? Why was the increase in pH much higher in the controls than for the other mesocosms. Obviously, some information is missing here. Do you know the reason why pH decreased so fast between day -4 and day -3? 2) Timing. The second concern I have is related to the division of the experiment in 2 phases. Phase 1 corresponds to the development of the diatom bloom extended up to the depletion of nitrate (day 0 to 4) and Phase 2 corresponds to the declining phase of the bloom in the absence of detectable nitrate. Except that this is not really true since temperature increased the speed at which chla built-up and nutrients were consumed (this is not really mentioned in 3.2). At 15°C, except for 1 mesocosm, nitrate was exhausted already on day 2 while at 10°C, NO3 in most mesocosms were actually exhausted on day 4. My point is that since T modified the timing of the bloom (and

its decline), it does not seem correct to me to consider fixed periods. The build-up of chla and all related statistical analyses should be conducted at 15°C between day 0 and 2, and all tests related to the decline of the bloom between day 3 and 13. Would that change some of your results? 3) Grazing. I regret that potentially the most exciting result of this experiment suggesting that pCO2 "positive" effects on phytoplankton were actually masked by significant increases in micro-grazing is not more developed. I understand the politics behind the publication of papers from a joint experiment, it would just bring much more value to your paper if these results were incorporated and discussed. Top-down control is very often neglected in these OA-OW experiments...

**Minor comments.**

L217: concentrations were L225: "suggesting a faster loss of pigments...". Not really convinced by that... Is the slope different? L230: "The strong correlation" I do not understand this sentence. How a correlation can suggest anything about importance? Figure 1a: label pHT in situ, why in situ?

СЗ

---

## Author Comment (AC1) · 25 Apr 2018

Referee 1 comments: The manuscript by Benard et al. describes the results from a mesocosm experiment that was designed to investigate the responses of a natural phytoplankton community to warming and acidification. The authors observed a clear stimulation of phytoplankton growth by temperature whereas acidification had no or only a minor effect. Although many experimental studies have been conducted in recent years to investigate phytoplankton responses to warming or to acidification little is known about combined effects. The data provided by this study are thus potentially valuable and interesting. However, important information is lacking in the current version and need to be included and discussed to improve the value of this manuscript. The set-up of the experiment was designed to keep pH in the acidified mesocosms constant, yielding a decrease of pH after the bloom. This differs to the natural environment where a phytoplankton bloom can substantially modify (increase) pH. It also differs to the earlier mesocosms experiments that the authors reference in their discussion. I suggest that the authors discuss implications of the differences in the set-up of experiments.

Author's response to general comments: We thank the reviewer for the thorough evaluation of the manuscript and the insightful comments. The comment on our experimental approach is very appropriate and, as suggested, a new paragraph was added where its implications are discussed (see below). The experimental protocol where pH and pCO2 are kept constant during the full experiment is indeed different to what would happen in nature where changes in photosynthesis and respiration during the bloom development would affect theses parameters. The main reason why these conditions were kept constant was to allow us to precisely measure and maximise inference of the effects of pH and pCO2 on different processes (e.g. phytoplankton photosynthesis in this paper, and dimethylsulfide production in a companion paper to be submitted) taking place during all phases of the bloom.

The following section has been added in the new version of the manuscript: 4.5: Implications and limitations During our study, we chose to keep the pH constant during the whole experiment instead of allowing it to vary with changes in photosynthesis and respiration during the bloom phases. This approach differs from previous mesocosm experiments where generally no subsequent CO2 manipulations are conducted after the initial targets are attained (Schulz et al. 2017 and therein). Keeping the pH and pCO2 conditions stable during our study allowed us to precisely quantify the effect of the changing pH/pCO2 on the processes taking place during the different phases of the bloom. Such control was not exercised in two of our mesocosms (i.e. the drifters). In these two mesocosms, the pH increased from 7.9 to 8.3 at 10°C, and from 7.9 to 8.7 at

15°C. Since the buffer capacity of acidified waters diminishes with increasing CO2, the drift in pCO2 and pH due to biological activity would have been even greater in the more acidified treatments (Delille et al., 2005; Riebesell et al., 2007). Hence, allowing the pH to drift in all mesocosms would have likely ended in an overlapping of the treatments where acidification effects would have been harder to detect. Thus, our experiment could be considered as an intermediate between strictly controlled small scale laboratory experiments and large scale pelagic mesocosm experiments in which only the initial conditions are set. By limiting pCO2 decrease under high CO2 drawdown due to photosynthesis during the bloom phase, we minimise confounding effects of pCO2 potentially overlapping in association with high biological activity in the mesocosms. Hence, the experimental conditions could be considered as extreme examples of acidification conditions, due to the extent of pCO2 values studied. However, the absence of OA effects on most biological parameters measured during our study, even under these extreme conditions, strengthens the argument that the phytoplankton community in LSLE is resistant to OA.

Referee comment: Nutrient concentration and irradiance are main factors controlling phytoplankton growth in seawater. The authors should asses how these factors may have affected cell growth and primary production. This includes: 1. Add a drawing of the set-up and placement of mesocosms and treatments within the container. Containers often bear the risk of self-shading, which would need to be considered.

Author's response: We thank the reviewer for pointing that out. Light is indeed an important driver of phytoplankton photosynthesis and growth. As requested, the following information has been added in the text as well as a new figure:

Modification (line 88) Old sentence: Each enclosure is sealed with a Plexiglas cover allowing the transmission of 90

New sentence: The mesocosms exhibit opaque walls and all lie on the same plane level as not to shade each other. Light penetrates the mesocosms only through a sealed

Plexiglas circular cover at their uppermost part. The cover allows the transmission of 90

New figure (1): Mesocosm setup

In terms of the impact of nutrient concentrations on cell growth and primary production, we discuss this further: please refer to the specific comment about line 211 below.

Referee comment: Please give absolute values for irradiance instead of

Author's response: We acknowledge that light intensity has the potential to affect phytoplankton response to acidification and warming. Light intensity varied between days over the duration of the experiment (see new figure below). We cannot exclude the possible interaction of light intensity with acidification or warming, or the effect the varying intensity has on the dynamics of primary production. However, as all mesocosms were subjected to the same fluctuating natural irradiance, and that pCO2 and temperature were the only factors changing between the treatments, we limited our interpretation to these two parameters.

New sentence (line 112): Incident light was variable during our experiment, with only few sunny days (Fig. 2).

New figure (2): Irradiance

Referee comment: It is important to consider that net primary production was measured. Hence, responses to warming and acidification may not only be related to photosynthetic production but also to respiration processes. Please discuss.

Author's response: We agree with the reviewer. The text was modified accordingly (Line 394).

Old sentence: The warming-induced decrease in carbon fixation measured during Phase II may thus result from an increase in respiration by the nitrogen-limited diatoms.

New sentence: The warming-induced decrease in carbon fixation measured during

Phase II may thus result from an increase in respiration by the nitrogen-limited diatoms during periods of darkness of the incubations.

Referee comment: Given that the authors did not add nutrients to the natural seawater, the strong increase in biomass (from 10 to up to 30 $\mu$g/L Chl a in one day) after incubation is very surprising. What could have limited phytoplankton growth in situ? Please discuss.

Author's response: The following phrases were added to address this phenomenon (line 302). New sentence: In situ nutrient conditions prior to the water collection were favourable for a bloom development. Based on previous studies, in situ phytoplankton growth was probably limited by light due to water turbidity and vertical mixing at the time of water collection (Levasseur et al. 1984). Grazing may also have played a role in keeping the in situ biomass of flagellates low prior to our sampling. However, a natural diatom fall bloom was observed in the days following the water collection in the adjacent region (Ferreyra, pers. comm.). The increased stability within the mesocosms, combined with the reduction of the grazing pressure (filtration on 250 $\mu$m) likely contributed to the fast accumulation of phytoplankton biomass.

Referee comment: There was a strong drop in pH prior to the acidification treatment on day -3. What may have been the reason for this drop?

Author's response (AR): The following modifications have been made to address the pH drop / pCO2 rise at the onset of the experiment.

Modification (line 200) Old sentence: The pH remained relatively stable throughout the experiment in the pH-controlled treatments, but decreased slightly during Phase II by an average of -0.14 $\pm$ 0.07 units relative to the target pHT (Fig. 1a).

New sentence: Following the filling of the mesocosms, the pHT in all mesocosms decreased from an average of 7.84 to 7.53. Throughout the rest of the experiment after treatments were applied, the pH remained relatively stable in the pH-controlled

treatments, but decreased slightly during Phase II by an average of -0.14 ± 0.07 units relative to the target pHT (Fig. 1a).

Addition (line 294) New sentence: The onset of the experiment was marked by an increase of $pCO_2$ on the day following the filling of the mesocosms. This phenomenon often takes place at the beginning of such experiments when pumping tends to break phytoplankton cells and larger debris into smaller ones. We attribute the rapid fluctuations in $pCO_2$ to the release of organic matter following the filling of the mesocosms with a stimulating effect on heterotrophic respiration, and hence $CO_2$ production.

Specific comments:

Line 87: add diameter and height of mesocosms AR: Line 87 has been modified as follow: (note that the dimensions of the mesocosms are now also presented in the new figure 1):

Old sentence: The mesocosms are cylindrical with a cone-shaped bottom within which mixing is achieved using a propeller fixed near the top.

New sentence: The mesocosms are cylindrical (2.67 m × 1.40 m) with a cone-shaped bottom within which mixing is achieved using a propeller fixed near the top (Fig. 1).

Line 99ff: add total duration of the experiment to the description AR: Line 100 has been modified as follow:

Old sentence: The water was collected at 5 m depth near Rimouski harbour (48° 28' 39.9" N, 68° 31' 03.0" W) on the 27th of September 2014. In situ conditions were: salinity = 26.52, temperature = 10 °C, nitrate ($NO_3-$) = 12.8 ± 0.6 $\mu$mol L-1, silicic acid ($Si(OH)4$) = 16 ± 2 $\mu$mol L-1, and soluble reactive phosphate (SRP) = 1.4 ± 0.3 $\mu$mol L-1. The same day (indicated as day -5 hereafter), the water was filtered through a 250 $\mu$m mesh while simultaneously filling the 12 mesocosm tanks by gravity with a custom made 'octopus' tubing system.

New sentence: The water was collected at 5 m depth near Rimouski harbour (48°

28' 39.9" N, 68° 31' 03.0" W) on the 27th of September 2014 (indicated as day -5 hereafter), and the experiment lasted until the 15th of October 2014 (day 13). In situ conditions were: salinity = 26.52, temperature = 10 °C, nitrate (NO3-) = 12.8 ± 0.6 $\mu$mol L-1, silicic acid (Si(OH)4) = 16 ± 2 $\mu$mol L-1, and soluble reactive phosphate (SRP) = 1.4 ± 0.3 $\mu$mol L-1. On day -5, the water was filtered through a 250 $\mu$m mesh while simultaneously filling the 12 mesocosm tanks by gravity with a custom made 'octopus' tubing system.

Line 104: give value for initial pCO2 AR: Line 104 has been modified as follow: Old sentence: The initial in situ temperature of 10 °C was maintained in the twelve mesocosms for the first 24 h (day -4).

New sentence: The initial pCO2 was 623 ± 7 $\mu$atm and the in situ temperature of 10 °C was maintained in the twelve mesocosms for the first 24 h (day -4).

Line 113ff: Add total amount of volume sampled from the mesocosms each day AR: The following was added to line 116: "Total amount of volume sampled every day was 24 liters or less."

Line 166: Was the no replicate incubation? Was the error within treatment assessed? AR: There was no replication of incubations. The number of bottles to handle was already quite extensive and the maximum capacity of our incubators had been reached. We chose to adopt the strategy of an increased number of treatments (mesocosms), which, even with reduced replication, allows greater power to characterise the functional relationships between OA parameters and organism or ecosystem response (Riebesell et al. 2011 (Guide to Best Practices in Ocean Acidification)). However, we conducted independent measures of particulate PP, dissolved PP, as well as total PP every day in all the mesocosms allowing us to verify that the fractions measured in particulate and dissolved Pp reliably added up to the total PP.

Line 171: Give irradiance values AR: The daily irradiances are now presented in Figure B.

Line 202: pCO2 was 1340 $\pm$ 150 $\mu$atm on day -3; why was the value so high? AR: The pCO2 doubled after the filling of the mesocosms most probably due to an increase in CO2 production following the release of organic matter and the increase in heterotrophic respiration. We conclude that the filling of the mesocosms tends to break phytoplankton cells and larger debris into smaller ones, with a stimulating effect on bacteria. Refer to additions on lines 200 and 294.

Line 211:' The three nutrients displayed a similar temporal depletion pattern following the development of the phytoplanktonic bloom.' I disagree the nutrients in the warm treatments were clearly reduced much faster. AR: Right. We meant that the general pattern was similar between the three nutrients (nitrate, soluble reactive phosphate and silicic acid) within each of the mesocosms, however we agree that clarity could be added here. We rephrased this part of the results section:

Old sentence: The three nutrients displayed a similar temporal depletion pattern following the development of the phytoplanktonic bloom.

New sentences: Within individual mesocosms, concentrations of nitrate, silicic acid and soluble reactive phosphate displayed similar temporal patterns following the development of the phytoplankton bloom. Overall, nutrient depletion was reached within 5 days in all mesocosms at 10°C, exception made of the drifter which became nutrient-deplete by day 3. Nutrient depletion was reached slightly earlier within the 15oC mesocosms, all of them displaying exhaustion within 3 days of the experiment. Accordingly, bloom development and primary production within each mesocosm were eventually limited by the supply in nutrients, irrespective of the temperature or pH treatment.

Line 217: 'Chl a concentrations where below 1 $\mu$g L-1 just after the filling of the mesocosms, and averaged 5.9 $\pm$ 0.6 $\mu$g L-1 on day 0' If Chla increased that much regardless of treatment; light limitation or exclusion of zooplankton probably had a major influence of phytoplankton development and should be considered in more depth. AR: Water for our experiments was collected near shore where turbidity is high in this part of the St.

Lawrence Estuary. We thus attribute the rapid response observed at the beginning of our incubations to an increase in light availability. The presence of high nutrient levels in the water at the beginning of the experiment also suggests that light intensity in the upper mixed layer was too low to allow the development of the bloom near the dock where the water was collected. Please refer to modifications made for line 302.

Line 327: The citation of Bach et al 2017 is not accurate as that study didn't determine carbon fixation AR: The citation of Bach et al., 2017 has been removed.

Figure 3: Wasn't the Chl a accumulation (day 0 to Chl a max) not much higher in the warm control? AR: Figure 3 y-axis has been adjusted.

Figure 5 g, f: same axis labelling different figure. . .please check. AR: Figure 5g, e, f labelling has been verified. Figure 5j, k, l label has been modified from "Chl a-normalized PP ($\mu$mol C ($\mu$g Chl a)-1 d-1)" to: "Chl a-normalized PD ($\mu$mol C ($\mu$g Chl a)-1 d-1)".

―――――――――――――――――――――――

[Figure]

**Fig. 1.**

[Figure]

**Fig. 2.**

---

## Author Comment (AC2) · 25 Apr 2018

Referee comments #2: The manuscript of Bénard and collaborators reports on an experiment that has been conducted using indoor mesocosms (2.6 m3) to test for the effect of ocean acidification and warming on the development of a fall phytoplankton bloom in the Lower St. Lawrence Estuary. The experiment setup comprised 2 sets of 6 mesocosms installed in two temperature-controlled containers, that were filled with seawater sieved onto 250 microns. In one container, the water temperature was raised by 5âŮęC compared to the mesocosms installed in the other container (10 vs. 15âŮęC). A gradient approach (no replicates) has been considered for pCO2/pH covering a range of pH from 7.2 to 8.6. The experiment lasted 13 days and covered the development of a bloom and its decline. Major conclusions of this study are that pCO2 has no effect on all measured parameters and processes while increasing temperature led to a faster build-up of chlorophyll and higher particulate primary production rates. Overall, this is a very well written manuscript that deals with an important topic. The introduction is well documented and shows that while this topic is of great importance, a fair amount of studies has already been conducted, including studies using in situ mesocosms in various environments. Although I would like to ultimately recommend this manuscript for publication in BG, I am concerned by 3 major aspects of this work and would like the authors to answer these comments.

Author's response: We would like to thank the reviewer for the general evaluation of the manuscript and the insightful comments. We will further discuss the following comments from the reviewer.

Referee comment: Realism. The authors clearly mention that the surface mixed-layer pCO2 is strongly modulated by biological productivity, yet they decided to run an experiment during which a bloom is produced and where carbonate chemistry has been maintained as constant. This would be acceptable if well explained and discussed, but the problem is that "control" mesocosms were actually not controlled (consider changing their name. . .) and pH was left increasing while the bloom was forming to (what I consider to be) very high and potentially unrealistic pH (?) values of 8.6. In situ pH was apparently close to 7.8, these "control" mesocosms appear to me as "perturbed"! Besides this major concern, I have to admit I do not understand how carbonate chemistry was controlled. The authors mention that "acidification" was carried out over day -1. On that day, I actually also observe a sudden increase of pH for the "controls", pH8 and pH7.8. . . How did that happen? Naturally? Why was the increase in pH much higher in the controls than for the other mesocosms. Obviously, some information is missing here. Do you know the reason why pH decreased so fast between day -4 and day -3?
Author's response: First, following this comment, the "Controls" have been more appropriately renamed "Drifters" to clearly show that the pCO2 in these specific mesocosms was not controlled. We noted that Reviewer #1 also pointed out the shortcomings in the discussion of the different approaches to control pH during this type of experiment.

The following section has been added in the new version of the manuscript: 4.5: Implications and limitations During our study, we chose to keep the pH constant during the whole experiment instead of allowing it to vary with changes in photosynthesis and respiration during the bloom phases. This approach differs from previous mesocosm experiments where generally no subsequent CO2 manipulations are conducted after the initial targets are attained (Schulz et al. 2017 and therein). Keeping the pH and pCO2 conditions stable during our study allowed us to precisely quantify the effect of the changing pH/pCO2 on the processes taking place during the different phases of the bloom. Such control was not exercised in two of our mesocosms (i.e. the drifters). In these two mesocosms, the pH increased from 7.9 to 8.3 at 10°C, and from 7.9 to 8.7 at 15°C. Since the buffer capacity of acidified waters diminishes with increasing CO2, the drift in pCO2 and pH due to biological activity would have been even greater in the more acidified treatments (Delille et al., 2005; Riebesell et al., 2007). Hence, allowing the pH to drift in all mesocosms would have likely ended in an overlapping of the treatments where acidification effects would have been harder to detect. Thus, our experiment could be considered as an intermediate between strictly controlled small scale laboratory experiments and large scale pelagic mesocosm experiments in which only the initial conditions are set. By limiting pCO2 decrease under high CO2 drawdown due to photosynthesis during the bloom phase, we minimise confounding effects of pCO2 potentially overlapping in association with high biological activity in the mesocosms. Hence, the experimental conditions could be considered as extreme examples of acidification conditions, due to the extent of pCO2 values studied. However, the absence of OA effects on most biological parameters measured during our study, even under these extreme conditions, strengthens the argument that the phytoplankton community in LSLE is resistant to OA.
To further clarify how the acidification and pH treatments were controlled, the following phrases have been added.

Addition (line 112): To attain initial targeted pH, CO2-satured artificial seawater was precisely added via an automatic delivery system to mesocosms M1 (7.4), M3 (7.6), M5 (7.2), M7 (7.4), M8 (7.2), and M10 (7.6). Mesocosms M2 (8.0), M4 (7.8), M6 (Drifter), M9 (8.0), M11 (Drifter) and M12 (7.8) were gently mixed to allow the outward degassing of the supersaturated CO2. Once the mesocosms had reached their target pH, the automatic system controlled the sporadic addition of CO2-saturated water to refrain the pH from rising. Only the "Drifters" were not controlled throughout the experiment.

Referee comment: Timing. The second concern I have is related to the division of the experiment in 2 phases. Phase 1 corresponds to the development of the diatom bloom extended up to the depletion of nitrate (day 0 to 4) and Phase 2 corresponds to the declining phase of the bloom in the absence of detectable nitrate. Except that this is not really true since temperature increased the speed at which chla built-up and nutrients were consumed (this is not really mentioned in 3.2). At 15âUeC, except for 1 mesocosm, nitrate was exhausted already on day 2 while at 10âUeC, NO3 in most mesocosms were actually exhausted on day 4. My point is that since T modified the timing of the bloom (and its decline), it does not seem correct to me to consider fixed periods. The build-up of chla and all related statistical analyses should be conducted at 15âUeC between day 0 and 2, and all tests related to the decline of the bloom between day 3 and 13. Would that change some of your results?

Author's response: Dividing the experiment into phases allows the disentanglement of the potential impacts caused by different processes and conditions occurring during different phases of a bloom, essentially its development and its decline. This is a strategy commonly used in published studies that delve into the impacts of OA on bloom dynamics. We previously considered different division criteria for the experiment (day of nitrate exhaustion, maximum ChI a concentration, averaged day of nitrate exhaustion) and ultimately opted for the averaged day of nitrate exhaustion as to mark the end
of the nutrient-rich development phase. This would allow comparisons with numerous mesocosm experiments that also divide their experiment using fixed periods. However, in many of those studies, the distinction between the phases was sharply defined and timing was not such a significant factor. In our study, the onset, peak Chl a buildup and decline of the blooms, showed variation, and overall timing of the blooms was different between temperature treatments. Thus, we agree with the reviewer's suggestion to modify the phase criteria and suggest to take it one step further to strengthen the inference of treatment effects. Assigning phase durations based on differential Chl a buildup between temperature treatments as the reviewer suggests (Phase I: days 0-4 at 10°C, and days 0-2 at 15°C) would exclude some data from mesocosms that are still in the growth phase from the analyses of that phase. For example, M3 and M5 maximum Chl a concentrations are attained on day 7, and M7 maximum Chl a concentration is achieved on day 4. Therefore, we suggest modifying the phases for each mesocosm as follow: Phase I (day 0 to day of maximum Chl a concentration) and Phase II (day after maximum Chl a concentration to day 13). By doing so, all the analyses on Phase I will be constrained to the Chl a accumulation phase for each mesocosm, while Phase II will be an accurate representation of the individual declining phases. This modification does not change the global narrative or conclusions of the manuscript but does carry a few modifications in the statistical outputs. The absence of acidification effects is still valid for all parameters measured, as they stand currently, except for picocyanobacterial abundance at 15°C during Phase II which shows a negative linear trend with increasing pCO2 using the new phase criteria. We already suggest in the paper that potential heightened grazing pressure could counteract the stimulating effect of increased COÂň2 availability on picocyanobacteria, and this is still valid. With regards to the temperature effects, the differences on the mean concentrations of Chl a would no longer be significant in either phases. However, the accumulation rate of Chl a, a parameter that better defines bloom development, is still significantly higher at 15°C, reflecting the faster accumulation of Chl a. The temperature effects on particulate primary production during the specific phases are no longer apparent, however our initial

**BGD**
conclusion that the PP is not affected over the full duration of the experiment remains valid. This will strengthen the conclusion that only the timing of the bloom development is affected by temperature, with negligible effects on the other parameters. Since we had already processed data in this manner and final figures can be swiftly produced to reflect the changes in the statistical analyses, we are confident that these modifications can bolster the paper and its findings.

Referee comment: Grazing. I regret that potentially the most exciting result of this experiment suggesting that pCO2 "positive" effects on phytoplankton were actually masked by significant increases in micro-grazing is not more developed. I understand the politics behind the publication of papers from a joint experiment, it would just bring much more value to your paper if these results were incorporated and discussed. Top-down control is very often neglected in these OA-OW experiments. . .

Author's response: We agree with this comment. The impact of the different treatments on zooplankton abundance will be discussed in a companion paper by colleagues.

Minor comments

L217: concentrations were AR: "Concentrations where" changed to "concentrations were"

L225: "suggesting a faster loss of pigments. . .". Not really convinced by that. . . Is the slope different? AR: Following the changes made with regards to phase criteria and ensuing statistical analyses, this section would be adjusted as follows:

Old section (line 223-226): During Phase II, we observed no significant effect of increasing pCO2 on the mean ChI a concentrations at the two temperatures tested. Nevertheless, during that phase, the mean ChI a concentrations decreased from 18.2  $\pm$  0.9  $\mu$ g L-1 at 10 °C to 12.4  $\pm$  0.7  $\mu$ g L-1 at 15 °C, suggesting a faster loss of the pigments following the depletion of NO3-.

New sentence: During Phase II, we observed no significant effect of increasing

BGD
pCOÂň2, nor temperature, on the mean ChI a concentrations following the depletion of NO3-.

L230: "The strong correlation" I do not understand this sentence. How a correlation can suggest anything about importance? AR: The sentence has been removed.

Figure 1a: label pHT in situ, why in situ? AR: All pHTÂÿare measured at 25°C and are computed to the temperatures of the mesocosms. The label "pHT in situ" meant that the pHT is calculated at the in situ temperature of each mesocosm. Therefore, for mesocosms M1–M6 the pHT is computed at 10°C, while for mesocosms M7–M12 the pHT is computed at 15°C. To avoid confusion, we changed the label to "pHT".

---

## Author Response (AR1)

**Referee #1 comments**: The manuscript by Benard et al. describes the results from a mesocosm experiment that was designed to investigate the responses of a natural phytoplankton community to warming and acidification. The authors observed a clear stimulation of phytoplankton growth by temperature whereas acidification had no or only a minor effect. Although many experimental studies have been conducted in recent years to investigate phytoplankton responses to warming or to acidification little is known about combined effects. The data provided by this study are thus potentially valuable and interesting. However, important information is lacking in the current version and need to be included and discussed to improve the value of this manuscript. The set-up of the experiment was designed to keep pH in the acidified mesocosms constant, yielding a decrease of pH after the bloom. This differs to the natural environment where a phytoplankton bloom can substantially modify (increase) pH. It also differs to the earlier mesocosms experiments that the authors reference in their discussion. I suggest that the authors discuss implications of the differences in the set-up of experiments.

**Author's response to general comments**: We thank the reviewer for the thorough evaluation of the manuscript and the insightful comments. The comment on our experimental approach is very appropriate and, as suggested, a new paragraph was added where its implications are discussed (see below). The experimental protocol where pH and $pCO_2$ are kept constant during the full experiment is indeed different to what would happen in nature where changes in photosynthesis and respiration during the bloom development would affect theses parameters. The main reason why these conditions were kept constant was to allow us to precisely measure and maximise inference of the effects of pH and $pCO_2$ on different processes (e.g. phytoplankton photosynthesis in this paper, and dimethylsulfide production in a companion paper to be submitted) taking place during all phases of the bloom.

The following section has been added in the new version of the manuscript:

**4.5: Implications and limitations**

During our study, we chose to keep the pH constant during the whole experiment instead of allowing it to vary with changes in photosynthesis and respiration during the bloom phases. This approach differs from previous mesocosm experiments where generally no subsequent $CO_2$ manipulations are conducted after the initial targets are attained (Schulz et al. 2017 and therein). Keeping the pH and $pCO_2$ conditions stable during our study allowed us to precisely quantify the effect of the changing pH/$pCO_2$ on the processes taking place during the different phases of the bloom. Such control was not exercised in two of our mesocosms (i.e. the drifters). In these two mesocosms, the pH increased from 7.9 to 8.3 at 10°C, and from 7.9 to 8.7 at 15°C. Since the buffer capacity of acidified waters diminishes with increasing $CO_2$, the drift in $pCO_2$ and pH due to biological activity would have been even greater in the more acidified treatments (Delille et al., 2005; Riebesell et al., 2007). Hence, allowing the pH to drift in all mesocosms would have likely ended in an overlapping of the treatments where acidification effects would have been harder to detect. Thus, our experiment could be considered as an intermediate between strictly controlled small scale laboratory experiments and large scale pelagic mesocosm experiments in which only the initial conditions are set. By limiting $pCO_2$ decrease under high $CO_2$ drawdown due to photosynthesis during the bloom phase, we minimise confounding effects of pCO$_2$ potentially overlapping in association with high biological activity in the mesocosms. Hence, the experimental conditions could be considered as extreme examples of acidification conditions, due to the extent of pCO$_2$ values studied. However, the absence of OA effects on most biological parameters measured during our study, even under these extreme conditions, strengthens the argument that the phytoplankton community in LSLE is resistant to OA.

**Referee comment:** Nutrient concentration and irradiance are main factors controlling phytoplankton growth in seawater. The authors should asses how these factors may have affected cell growth and primary production. This includes: 1. Add a drawing of the set-up and placement of mesocosms and treatments within the container. Containers often bear the risk of self-shading, which would need to be considered.

**Author's response:** We thank the reviewer for pointing that out. Light is indeed an important driver of phytoplankton photosynthesis and growth. As requested, the following information has been added in the text as well as a new figure:

**Modification (line 88)**

Old sentence:  Each enclosure is sealed with a Plexiglas cover allowing the transmission of 90 % of photosynthetically active radiation (PAR; 400–700 nm), 50–85 % of solar UVB (280–315 nm) and 85–90 % of UVA (315–400 nm).

New sentence:  The mesocosms exhibit opaque walls and all lie on the same plane level as not to shade each other. Light penetrates the mesocosms only through a sealed Plexiglas circular cover at their uppermost part.  The cover allows the transmission of 90 % of photosynthetically active radiation (PAR; 400–700 nm), 85–90 % of UVA (315–400 nm), and 50–

85 % of solar UVB (280–315 nm).

New figure (A):

[Figure]

In terms of the impact of nutrient concentrations on cell growth and primary production, we discuss this further: please refer to the specific comment about line 211 below.

**Referee comment:** Please give absolute values for irradiance instead of % shading. What was the light:dark cycle? Since primary production measurements were carried out over 24h, some incubation hours will have been in the dark if natural sunlight was applied. It is important to inform about the potential role of dark respiration during the incubations. Since light intensity has been shown to co-affect phytoplankton responses to acidification and warming, it is absolutely necessary to show and discuss the light climate in mesocosms and incubations.

**Author's response:**
We acknowledge that light intensity has the potential to affect phytoplankton response to acidification and warming. Light intensity varied between days over the duration of the experiment (see new figure below). We cannot exclude the possible interaction of light intensity with acidification or warming, or the effect the varying intensity has on the dynamics of primary production. However, as all mesocosms were subjected to the same fluctuating natural irradiance, and that $pCO_2$ and temperature were the only factors changing between the treatments, we limited our interpretation to these two parameters.

New sentence (line 112): Incident light was variable during our experiment, with only few sunny days (Fig. B).

New figure (B):

[Figure]

**Referee comment**: It is important to consider that net primary production was measured. Hence, responses to warming and acidification may not only be related to photosynthetic production but also to respiration processes. Please discuss.

**Author's response:**

We agree with the reviewer. The text was modified accordingly (Line 394).

Old sentence: The warming-induced decrease in carbon fixation measured during Phase II may thus result from an increase in respiration by the nitrogen-limited diatoms.

New sentence: The warming-induced increase in fixed carbon being release in the dissolved fraction likely stems from increased exudation by phytoplankton, or sloppy feeding / excretion following ingestion by grazers (Kim et al., 2011). The increase in fixed carbon released as dissolved organic carbon (DOC) measured during Phase II may also result from greater respiration by the nitrogen-limited diatoms during periods of darkness of the incubations, as dark phytoplankton respiration rates generally increase with temperature (Butrón et al., 2009; Robarts and Zohary, 1987). Moreover, the enclosures do not permit the sinking and export of particulates organic carbon (POC), allowing a further transformation into DOC by heterotrophic bacteria, a process that could be exacerbated under warming (Wohlers et al., 2009).

**Referee comment:** Given that the authors did not add nutrients to the natural seawater, the strong increase in biomass (from 10 to up to 30 µg/L Chl a in one day) after incubation is very surprising. What could have limited phytoplankton growth in situ? Please discuss.

**Author's response:**

The following phrases were added to address this phenomenon (line 302).

New sentence: In situ nutrient conditions prior to the water collection were favourable for a bloom development. Based on previous studies, in situ phytoplankton growth was probably limited by light due to water turbidity and vertical mixing at the time of water collection (Levasseur et al. 1984). Grazing may also have played a role in keeping the in situ biomass of flagellates low prior to our sampling. However, a natural diatom fall bloom was observed in the days following the water collection in the adjacent region (Ferreyra, pers. comm.). The increased stability within the mesocosms, combined with the reduction of the grazing pressure (filtration on 250 µm) likely contributed to the fast accumulation of phytoplankton biomass.

**Referee comment:** There was a strong drop in pH prior to the acidification treatment on day -3. What may have been the reason for this drop?

**Author's response (AR):**

The following modifications have been made to address the pH drop / $pCO_2$ rise at the onset of the experiment.

Modification (line 200)

Old sentence: The pH remained relatively stable throughout the experiment in the pH-controlled treatments, but decreased slightly during Phase II by an average of -0.14 ± 0.07 units relative to the target $pH_T$ (Fig. 1a).

New sentence: Following the filling of the mesocosms, the $pH_T$ in all mesocosms decreased from an average of 7.84 to 7.53. Throughout the rest of the experiment after treatments were applied, the pH remained relatively stable in the pH-controlled treatments, but decreased slightly during Phase II by an average of -0.14 ± 0.07 units relative to the target $pH_T$ (Fig. 1a).

Addition (line 294)

New sentence: The onset of the experiment was marked by an increase of $pCO_2$ on the day following the filling of the mesocosms. This phenomenon often takes place at the beginning of such experiments when pumping tends to break phytoplankton cells and larger debris into smaller ones. We attribute the rapid fluctuations in $pCO_2$ to the release of organic matter following the filling of the mesocosms with a stimulating effect on heterotrophic respiration, and hence $CO_2$ production.

**Specific comments:**

**Line 87:** add diameter and height of mesocosms

**AR:** Line 87 has been modified as follow: (note that the dimensions of the mesocosms are now also presented in the new figure A):

Old sentence: The mesocosms are cylindrical with a cone-shaped bottom within which mixing is achieved using a propeller fixed near the top.

New sentence: The mesocosms are cylindrical (2.67 m × 1.40 m) with a cone-shaped bottom within which mixing is achieved using a propeller fixed near the top.

**Line 99ff:** add total duration of the experiment to the description

**AR:** Line 100 has been modified as follow:

Old sentence:
The water was collected at 5 m depth near Rimouski harbour (48° 28' 39.9" N, 68° 31' 03.0" W) on the 27th of September 2014. In situ conditions were: salinity = 26.52, temperature = 10 °C, nitrate $(NO_3^-)$ = 12.8 ± 0.6 µmol L$^{-1}$, silicic acid $(Si(OH)_4)$ = 16 ± 2 µmol L$^{-1}$, and soluble reactive phosphate (SRP) = 1.4 ± 0.3 µmol L$^{-1}$. The same day (indicated as day -5 hereafter), the water was filtered through a 250 µm mesh while simultaneously filling the 12 mesocosm tanks by gravity with a custom made 'octopus' tubing system.

New sentence:
The water was collected at 5 m depth near Rimouski harbour (48° 28' 39.9" N, 68° 31' 03.0" W) on the 27th of September 2014 (indicated as day -5 hereafter), and the experiment lasted until the 15th of October 2014 (day 13). In situ conditions were: salinity = 26.52, temperature = 10 °C, nitrate $(NO_3^-)$ = 12.8 ± 0.6 µmol L$^{-1}$, silicic acid $(Si(OH)_4)$ = 16 ± 2 µmol L$^{-1}$, and soluble reactive phosphate (SRP) = 1.4 ± 0.3 µmol L$^{-1}$. On day -5, the water was filtered through a 250 µm mesh while simultaneously filling the 12 mesocosm tanks by gravity with a custom made 'octopus' tubing system.

**Line 104:** give value for initial pCO2

**AR:** Line 104 has been modified as follow:

Old sentence: The initial in situ temperature of 10 °C was maintained in the twelve mesocosms for the first 24 h (day -4).

New sentence: The initial $pCO_2$ was 623 ± 7 µatm and the in situ temperature of 10 °C was maintained in the twelve mesocosms for the first 24 h (day -4).

**Line 113ff**: Add total amount of volume sampled from the mesocosms each day

**AR:** The following was added to line 116: "Total amount of volume sampled every day was 24 L or less."

**Line 166:** Was the no replicate incubation? Was the error within treatment assessed?

**AR:** There was no replication of incubations. The number of bottles to handle was already quite extensive and the maximum capacity of our incubators had been reached. We chose to adopt the strategy of an increased number of treatments (mesocosms), which, even with reduced replication, allows greater power to characterise the functional relationships between OA parameters and organism or ecosystem response (Riebesell et al. 2011 (Guide to Best Practices in Ocean Acidification)). However, we conducted independent measures of particulate $P_P$, dissolved $P_P$, as well as total $P_P$ every day in all the mesocosms allowing us to verify that the fractions measured in particulate and dissolved $P_p$ reliably added up to the total $P_P$.

**Line 171:** Give irradiance values

**AR:** The daily irradiances are now presented in Figure B.

**Line 202:** pCO2 was 1340 ± 150 µatm on day -3; why was the value so high?

**AR:** The $pCO_2$ doubled after the filling of the mesocosms most probably due to an increase in $CO_2$ production following the release of organic matter and the increase in heterotrophic respiration. We conclude that the filling of the mesocosms tends to break phytoplankton cells and larger debris into smaller ones, with a stimulating effect on bacteria. Refer to additions on lines 200 and 294.

**Line 211:**' The three nutrients displayed a similar temporal depletion pattern following the development of the phytoplanktonic bloom.' I disagree the nutrients in the warm treatments were clearly reduced much faster.

**AR:** Right. We meant that the general pattern was similar between the three nutrients (nitrate, soluble reactive phosphate and silicic acid) within each of the mesocosms, however we agree that clarity could be added here. We rephrased this part of the results section:

Old sentence: The three nutrients displayed a similar temporal depletion pattern following the development of the phytoplanktonic bloom.

New sentences: Within individual mesocosms, concentrations of nitrate, silicic acid and soluble reactive phosphate displayed similar temporal patterns following the development of the phytoplankton bloom. Overall, nutrient depletion was reached within 5 days in all mesocosms at 10°C, exception made of the drifter which became nutrient-deplete by day 3. Nutrient depletion was reached slightly earlier within the 15ºC mesocosms, all of them displaying exhaustion within 3 days of the experiment. Accordingly, bloom development and primary production within each mesocosm were eventually limited by the
supply in nutrients, irrespective of the temperature or pH treatment.
**Line 217:** 'Chl a concentrations where below 1 µg L-1 just after the filling of the mesocosms, and averaged 5.9 ± 0.6 µg L-1
on day 0' If Chla increased that much regardless of treatment; light limitation or exclusion of zooplankton probably had a
major influence of phytoplankton development and should be considered in more depth.
**AR:** Water for our experiments was collected near shore where turbidity is high in this part of the St. Lawrence Estuary. We
thus attribute the rapid response observed at the beginning of our incubations to an increase in light availability. The presence
of high nutrient levels in the water at the beginning of the experiment also suggests that light intensity in the upper mixed layer
was too low to allow the development of the bloom near the dock where the water was collected. Please refer to modifications
made for line 302.
**Line 327:** The citation of Bach et al 2017 is not accurate as that study didn't determine carbon fixation
**AR:** The citation of Bach et al., 2017 has been removed.
**Figure 3**: Wasn't the Chl a accumulation (day 0 to Chl a max) not much higher in the warm control?
**AR:** Figure 3 y-axis has been adjusted.
**Figure 5 g, f:** same axis labelling different figure. . .please check.
**AR:** Figure 5g, e, f labelling has been verified. Figure 5j, k, l label has been modified from "Chl $a$-normalized $P_P$ (µmol C (µg
Chl $a)^{-1}$ $d^{-1}$)" to: "Chl $a$-normalized $P_D$ (µmol C (µg Chl $a)^{-1}$ $d^{-1}$)".

**Referee comments #2**: The manuscript of Bénard and collaborators reports on an experiment that has been conducted using indoor mesocosms (2.6 m3) to test for the effect of ocean acidification and warming on the development of a fall phytoplankton bloom in the Lower St. Lawrence Estuary. The experiment setup comprised 2 sets of 6 mesocosms installed in two temperature-controlled containers, that were filled with seawater sieved onto 250 microns. In one container, the water temperature was raised by 5°C compared to the mesocosms installed in the other container (10 vs. 15°C). A gradient approach (no replicates) has been considered for pCO2/pH covering a range of pH from 7.2 to 8.6. The experiment lasted 13 days and covered the development of a bloom and its decline. Major conclusions of this study are that pCO2 has no effect on all measured parameters and processes while increasing temperature led to a faster build-up of chlorophyll and higher particulate primary production rates. Overall, this is a very well written manuscript that deals with an important topic. The introduction is well documented and shows that while this topic is of great importance, a fair amount of studies has already been conducted, including studies using in situ mesocosms in various environments. Although I would like to ultimately recommend this manuscript for publication in BG, I am concerned by 3 major aspects of this work and would like the authors to answer these comments.

**Author's response:** We would like to thank the reviewer for the general evaluation of the manuscript and the insightful comments. We will further discuss the following comments of the reviewer.

**Referee comment:** Realism. The authors clearly mention that the surface mixed-layer pCO2 is strongly modulated by biological productivity, yet they decided to run an experiment during which a bloom is produced and where carbonate chemistry has been maintained as constant. This would be acceptable if well explained and discussed, but the problem is that "control" mesocosms were actually not controlled (consider changing their name. . .) and pH was left increasing while the bloom was forming to (what I consider to be) very high and potentially unrealistic pH (?) values of 8.6. In situ pH was apparently close to 7.8, these "control" mesocosms appear to me as "perturbed"! Besides this major concern, I have to admit I do not understand how carbonate chemistry was controlled. The authors mention that "acidification" was carried out over day -1. On that day, I actually also observe a sudden increase of pH for the "controls", pH8 and pH7.8. . . How did that happen? Naturally? Why was the increase in pH much higher in the controls than for the other mesocosms. Obviously, some information is missing here. Do you know the reason why pH decreased so fast between day -4 and day -3?

**Author's response:** First, following this comment, the "Controls" have been more appropriately renamed "Drifters" to clearly show that the pCO$_2$ was not controlled. We noted that Reviewer #1 also pointed out the shortcomings in the discussion of the different approaches to control pH during this type of experiment.

The following section has been added in the new version of the manuscript:
**4.5: Implications and limitations**

During our study, we chose to keep the pH constant during the whole experiment instead of allowing it to vary with changes in photosynthesis and respiration during the bloom phases. This approach differs from previous mesocosm experiments where generally no subsequent $CO_2$ manipulations are conducted after the initial targets are attained (Schulz et al. 2017 and therein).

Keeping the pH and $pCO_2$ conditions stable during our study allowed us to precisely quantify the effect of the changing pH/$pCO_2$ on the processes taking place during the different phases of the bloom. Such control was not exercised in two of our mesocosms (i.e. the drifters). In these two mesocosms, the pH increased from 7.9 to 8.3 at 10°C, and from 7.9 to 8.7 at 15°C.

Since the buffer capacity of acidified waters diminishes with increasing $CO_2$, the drift in $pCO_2$ and pH due to biological activity would have been even greater in the more acidified treatments (Delille et al., 2005; Riebesell et al., 2007). Hence, allowing the pH to drift in all mesocosms would have likely ended in an overlapping of the treatments where acidification effects would have been harder to detect. Thus, our experiment could be considered as an intermediate between strictly controlled small scale laboratory experiments and large scale pelagic mesocosm experiments in which only the initial conditions are set. By limiting

$pCO_2$ decrease under high $CO_2$ drawdown due to photosynthesis during the bloom phase, we minimise confounding effects of

$pCO_2$ potentially overlapping in association with high biological activity in the mesocosms. Hence, the experimental conditions could be considered as extreme examples of acidification conditions, due to the extent of $pCO_2$ values studied. However, the absence of OA effects on the biological parameters measured during our study, even under these extreme conditions, strengthens the argument that the phytoplankton community in LSLE is resistant to OA.

To further clarify how the acidification and pH treatments were applied, the following has been added.

Addition (line 112):

To attain initial targeted pH, $CO_2$-satured artificial seawater was added to the mesocosms that needed a pH lowering while mesocosms M2 (8.0), M4 (7.8), M6 (Drifter), M9 (8.0), M11 (Drifter) and M12 (7.8) were openly mixed to allow the degassing of the supersaturated $CO_2$. Once the mesocosms had reached their target pH, the automatic system controlled the sporadic addition of $CO_2$-saturated water to refrain the pH from rising. Only the "Drifters" were not controlled throughout the experiment.

**Referee comment:** Timing. The second concern I have is related to the division of the experiment in 2 phases. Phase 1

corresponds to the development of the diatom bloom extended up to the depletion of nitrate (day 0 to 4) and Phase 2

corresponds to the declining phase of the bloom in the absence of detectable nitrate. Except that this is not really true since temperature increased the speed at which chla built-up and nutrients were consumed (this is not really mentioned in 3.2). At

15°C, except for 1 mesocosm, nitrate was exhausted already on day 2 while at 10°C, NO3 in most mesocosms were actually exhausted on day 4. My point is that since T modified the timing of the bloom (and its decline), it does not seem correct to me to consider fixed periods. The build-up of chla and all related statistical analyses should be conducted at 15°C between day 0

and 2, and all tests related to the decline of the bloom between day 3 and 13. Would that change some of your results?

**Author's response:**

We agree with the reviewer on the need for a better representation of the different phases in relation to the treatments but would previously like to inform on our decision to initially choose a fixed period. Firstly, it is imperative to divide the experiment in two phases not to confound effects caused by different processes and conditions during the development or the decline of the bloom. We previously considered multiple division method for the experiment (day of nitrate exhaustion, maximum Chl *a* concentration, averaged day of nitrate exhaustion) and ultimately opted for the averaged day of nitrate exhaustion as to mark the end of the nutrient-rich development phase. By choosing a single day as the divider for the phases, we could accurately compare our results with numerous mesocosms experiment that also divide their experiment using fixed periods. However, in many of those cases, the distinction between the phases were sharply defined as timing was not an important factor.

Thus, we agree with the reviewer suggestion but would like to take it one step further. Assigning phases duration based on temperature treatments as the reviewer suggests (Phase I: days 0-4 at 10°C, and days 0-2 at 15°C) would exclude some data from mesocosms that are still in the growth phase from the analyses of that phase, as M3 and M5 maximum Chl *a* concentrations are attained on day 7, and M7 maximum Chl *a* concentration is on day 4. Therefore, we suggest modifying the phases for each mesocosm as follow: Phase I (day 0 to day of maximum Chl *a* concentration) and Phase II (day after maximum Chl *a* concentration to day 13). By doing so, all the analyses on the Phase I will be constrained to the Chl *a* accumulation phase for each mesocosm, while Phase II will be an accurate representation of the individual declining phase. This modification carries some changes in the significance of the results of the analyses, although it does not change the overall narrative or conclusion of the manuscript. Namely, the absence of acidification effects is still valid for all parameters measured, except for picocyanobacteria abundance at 15°C during Phase II which is shows a negative linear trend with increasing $pCO_2$. We already suggested that increases in grazing pressures could counteract the stimulating effect of increased $CO_2$ availability on picocyanobacteria, and this is still valid. With regards to the temperature effects, the differences on the mean concentrations of Chl *a* is not significant in either phases. Although, the accumulation rate of Chl *a* is still higher at 15°C, reflecting the faster accumulation of Chl *a*. The temperature effects on particulate primary production during the both phases have also been discarded, yet our conclusion remains that the $P_P$ is not affected over the full duration of the experiment. This will strengthen the conclusion that only the timing of the bloom development is affected by temperature, with negligible effects on the other parameters.

Moreover, we had already processed data in this manner and final figures can easily be produced to reflect the changes in the statistical analyses.

**Referee comment:** Grazing. I regret that potentially the most exciting result of this experiment suggesting that pCO2 "positive" effects on phytoplankton were actually masked by significant increases in micro-grazing is not more developed. I understand the politics behind the publication of papers from a joint experiment, it would just bring much more value to your paper if these results were incorporated and discussed. Top-down control is very often neglected in these OA-OW experiments.
..

**Author's response:**

We agree with this comment. The impact of the different treatments on zooplankton abundance will be discussed in a
companion paper by colleagues.

**Minor comments**

**L217**: concentrations were
**AR:** "Concentrations where" changed to "concentrations were"

**L225**: "suggesting a faster loss of pigments. . .". Not really convinced by that. . . Is the slope different?
**AR:** Following the changes made with regards to the statistical analyses, this section was adjusted.

Old section (line 223-226):
During Phase II, we observed no significant effect of increasing $pCO_2$ on the mean Chl $a$ concentrations at the two temperatures
tested. Nevertheless, during that phase, the mean Chl $a$ concentrations decreased from $18.2 \pm 0.9 \, \mu g \, L^{-1}$ at 10 °C to
$12.4 \pm 0.7 \, \mu g \, L^{-1}$ at 15 °C, suggesting a faster loss of the pigments following the depletion of $NO_3^-$.

New sentence:
During Phase II, we observed no significant effect of $pCO_2$, temperature, and the interaction of those factors on the mean
Chl $a$ concentrations following the depletion of $NO_3^-$.
**L230**: "The strong correlation" I do not understand this sentence. How a correlation can suggest anything about importance?
**AR:** The sentence has been modified.

New sentence:
The correlation between the nanophytoplankton abundance and Chl $a$ ($r^2 = 0.75$, $p < 0.001$, df = 166) suggests that this
phytoplankton size class was responsible for most of the biomass build-up throughout the experiment.

**Figure 1a**: label pHT in situ, why in situ?

**AR:** All $pH_T$ are measured at 25°C and are computed to the temperatures of the mesocosms. The label "$pH_T$ in situ" meant
that the $pH_T$ is calculated at the in situ temperature of each mesocosm. Therefore, for mesocosms M1–M6 the $pH_T$ is computed
at 10°C, while for mesocosms M7–M12 the $pH_T$ is computed at 15°C. To avoid confusion, we changed the label to "$pH_T$".

[revised manuscript text omitted]
 150$ $\mu$atm on day -3, and ranging from 564 to 2902 $\mu$atm at 10 °C, and from 363 to 2884 $\mu$atm at 15 °C on day 0 following the acidification (Fig. 31b; Table 1). The pH$_T$ in the DrifterControls (M6 and M11) increased from 7.896 and 7.862 on day 0 at 10 °C and 15 °C, respectively, to 8.307 and 8.554 on day 13, reflecting the balance between CO$_2$ uptake and metabolic CO$_2$ production over the duration of the experiment. On the last day, pCO$_2$ in all mesocosms ranged from 186 to 3695 $\mu$atm at 10 °C, and from 90 to 3480 $\mu$atm at 15 °C. The temperature of the mesocosms in each container remained within $\pm$ 0.1 °C of the target temperature throughout the experiment and averaged $10.04 \pm 0.02$ °C for mesocosms M1 through M6, and $15.0 \pm 0.1$ °C for mesocosms M7 through M12 (Fig. 31c; Table 1).

**3.2 Dissolved inorganic nutrient concentrations**

Nutrient concentrations averaged $9.1 \pm 0.5$ $\mu$mol L$^{-1}$ for NO$_3^-$, $13.4 \pm 0.3$ $\mu$mol L$^{-1}$ for Si(OH)$_4$, and $0.91 \pm 0.03$ $\mu$mol L$^{-1}$ for SRP on day 0 (Fig. 31d, 
[revised manuscript text omitted]

| | Day of max Chl *a* | $pH_T$ | $pCO_2$ (µatm) | $pH_T$ | $pCO_2$ (µatm) | Temperature (°C) |
|---|---|---|---|---|---|---|
| M1 (7.4 – 10 °C) | 4 | 7.32 ± 0.01 | 2231 ± 25 | 7.28 ± 0.02 | 2437 ± 92 | 10.06 ± 0.01 |
| M2 (8.0 – 10 °C) | 4 | 7.84 ± 0.01 | 628 ± 16 | 7.74 ± 0.03 | 814 ± 65 | 10.00 ± 0.01 |
| M3 (7.6 – 10 °C) | 7 | 7.54 ± 0.01 | 1294 ± 18 | 7.48 ± 0.02 | 1503 ± 64 | 10.07 ± 0.01 |
| M4 (7.8 – 10 °C) | 4 | 7.71 ± 0.01 | 868 ± 13 | 7.66 ± 0.01 | 976 ± 29 | 10.04 ± 0.01 |
| M5 (7.2 – 10 °C) | 7 | 7.17 ± 0.01 | 3122 ± 35 | 7.15 ± 0.01 | 3315 ± 94 | 10.03 ± 0.01 |
| M6 (Drifter – 10 °C) | 4 | 7.93 ± 0.01 | 503 ± 15 | 8.22 ± 0.03 | 251 ± 25 | 10.02 ± 0.01 |
| M7 (7.4 – 15 °C) | 4 | 7.38 ± 0.01 | 2004 ± 44 | 7.31 ± 0.02 | 2399 ± 120 | 15.00 ± 0.01 |
| M8 (7.2 – 15 °C) | 2 | 7.21 ± 0.01 | 2961 ± 58 | 7.18 ± 0.01 | 3179 ± 74 | 15.01 ± 0.01 |
| M9 (8.0 – 15 °C) | 2 | 7.85 ± 0.01 | 454 ± 13 | 7.79 ± 0.02 | 545 ± 25 | 15.03 ± 0.01 |
| M10 (7.6 – 15 °C) | 2 | 7.54 ± 0.01 | 1364 ± 22 | 7.44 ± 0.02 | 1746 ± 106 | 14.94 ± 0.01 |
| M11 (Drifter – 15 °C) | 1 | 8.07 ± 0.01 | 388 ± 90 | 8.59 ± 0.02 | 84 ± 7 | 14.96 ± 0.02 |
| M12 (7.8 – 15 °C) | 2 | 7.67 ± 0.01 | 1001 ± 31 | 7.59 ± 0.01 | 1215 44± | 14.98 ± 0.02 |

**Table 12. Results of the generalized least squares models (gls) tests for the effects of temperature. pCO₂. and their interaction during**
**Phase I (day 0 to day of maximum Chl *a* concentrationday 4). Separate analysis with pCO₂ as a continuous factor were performed**
**when temperature had a significant effect. Chl *a* concentration, nanophytoplankton abundance, picoeukaryote abundance,**
**picocyanobacteria abundance, particulate and dissolved primary production, and Chl *a*-normalized particulate and dissolved**
**primary production. Significant results are in bold. *p < 0.05.**

| Response Variable | Factor | df | t-value | p-value |
|---|---|---|---|---|
| Mean Chl *a* concentration ($\mu$g L$^{-1}$) | TemperatureTemperature | 88 | 2.0042.3 07 | 0.0800.0 50* |
| | pCO₂pCO₂ (10 °C) | 84 | -0.464-1.362 | 0.6550.2 45 |
| | pCO₂ x TemperaturepCO₂ (15 °C) | 84 | 0.244-1.263 | 0.8130.2 75 |
| Mean nanophytoplankton abundance ($\times$ 10$^6$ cells L$^{-1}$) | TemperatureTemperature | 88 | 2.7252.9 80 | 0.026*0. 018* |
| | pCO₂ (10°C)pCO₂ (10 °C) | 44 | -2.285-2.729 | 0.0840.0 53 |
| | pCO₂ (15°C)pCO₂ (15 °C) | 44 | -1.191-1.231 | 0.2990.2 86 |
| Mean picoeukaryote abundance ($\times$ 10$^6$ cells L$^{-1}$) | TemperatureTemperature | 88 | 1.0561.1 57 | 0.3220.2 81 |
| | pCO₂pCO₂ | 88 | -1.159-1.070 | 0.2800.3 16 |
| | pCO₂ x TemperaturepCO₂ × Temperature | 88 | 1.1251.0 85 | 0.2930.3 09 |
| Mean picocyanobacteria abundance ($\times$ 10$^6$ cells L$^{-1}$) | TemperatureTemperature | 88 | 0.8913.0 66 | 0.3990.0 15* |
| | pCO₂pCO₂ (10 °C) | 84 | 0.9910.1 25 | 0.3510.9 07 |

Formatted Table

| Response | Factor | df | t | p |
|---|---|---|---|---|
| | pCO₂ x TemperaturepCO₂ (15 °C) | 84 | -1.166-2.268 | 0.2770.086 |
| | TemperatureTemperature | 88 | -0.1242.690 | 0.9050.028* |
| Particulate primary production (µmol C L⁻¹) | pCO₂pCO₂ (10 °C) | 84 | -1.011-1.617 | 0.3420.181 |
| | pCO₂ x TemperaturepCO₂ (15 °C) | 84 | 0.867-0.992 | 0.4110.378 |
| | TemperatureTemperature | 88 | -1.4290.756 | 0.1910.472 |
| Dissolved primary production (µmol C L⁻¹) | pCO₂pCO₂ | 88 | -0.569-0.901 | 0.5850.394 |
| | pCO₂ x TemperaturepCO₂ x Temperature | 88 | 0.7230.956 | 0.4900.367 |
| | TemperatureTemperature | 88 | 1.6892.592 | 0.1300.032* |
| Chl a-normalized particulate primary production (µmol C (µg Chl a)⁻¹ d⁻¹) | pCO₂pCO₂ (10 °C) | 84 | 0.107-1.467 | 0.9180.216 |
| | pCO₂ x TemperaturepCO₂ (15 °C) | 84 | -0.381-0.840 | 0.7130.448 |
| Chl a-normalized dissolved primary production (µmol C (µg Chl a)⁻¹ d⁻¹) | TemperatureTemperature | 88 | -1.046-0.350 | 0.3260.735 |
| | pCO₂pCO₂ | 88 | -0.381-0.397 | 0.7130.702 |

| pCO$_2$ x Temperature | 8 | 0.449  | 0.665  |
|---|---|---|---|

**Table 32. Results of the generalized least squares models (gls) tests for the effects of temperature, $pCO_2$ and their interaction.**
**Separate analysis with $pCO_2$ as a continuous factor were performed when temperature had a significant effect. Accumulation rate**
**of Chl *a* (day 0 to maximum Chl *a* concentration), maximum Chl *a* concentration, growth rate of nanophytoplankton (day 0 to**
**maximum nanophytoplankton abundance), and maximum nanophytoplankton abundance. Significant results are in bold. \*p < 0.05.**

| Response Variable | Factor | df | t-value | p-value |
|---|---|---|---|---|
| Accumulation rate of Chl *a* (day$^{-1}$) | Temperature | **8** | **2.679** | **0.028\*** |
| | $pCO_2$ (10 °C) | 4 | -1.476 | 0.214 |
| | $pCO_2$ (15 °C) | 4 | -1.759 | 0.154 |
| Maximum Chl *a* concentration (µg L$^{-1}$) | Temperature | 8 | 1.305 | 0.228 |
| | $pCO_2$ | 8 | -0.387 | 0.709 |
| | $pCO_2 \times$ Temperature | 8 | 0.022 | 0.983 |
| Growth rate of nanophytoplankton (day$^{-1}$) | Temperature | **8** | **2.534** | **0.035\*** |
| | $pCO_2$ (10 °C) | 4 | -0.882 | 0.403 |
| | $pCO_2$ (15 °C) | 4 | 0.601 | 0.564 |
| Maximum nanophytoplankton abundance ($\times 10^6$ cells L$^{-1}$) | Temperature | 8 | 1.380 | 0.205 |
| | $pCO_2$ | 8 | -0.735 | 0.484 |
| | $pCO_2 \times$ Temperature | 8 | 0.302 | 0.770 |

[Figure]

Table 4. Results of the generalized least squares models (gls) tests for the effects of temperature, $pCO_2$, and their interaction during Phase II (day _after maximum Chl a_ to day 13). Separate analysis with $pCO_2$ as a continuous factor were performed when temperature had a significant effect. Chl _a_ concentration, nanophytoplankton abundance, picoeukaryote abundance, picocyanobacteria abundance, particulate and dissolved primary production, and Chl _a_-normalized particulate and dissolved primary production. Significant results are in bold. *p < 0.05, **p < 0.01, ***p < 0.001.

| Response Variable | Factor | df | t-value | p-value |
|---|---|---|---|---|
| Mean Chl _a_ concentration ($\mu$g L$^{-1}$) | Temperature | 88 | -1.539  | 0.162  |
| | $pCO_2$  (10 °C) | 84 | 0.733  | 0.484  |
| | $pCO_2$ x Temperature  | 84 | 0.156  | 0.880  |
| Mean nanophytoplankton abundance ($\times 10^6$ cells L$^{-1}$) | Temperature | 88 | -0.528  | 0.612  |
| | $pCO_2$  | 88 | 1.264  | 0.242  |
| | $pCO_2$ x Temperature  | 88 | 0.699  | 0.505  |
| Mean picoeukaryotes abundance ($\times 10^6$ cells L$^{-1}$) | Temperature | 88 | 1.628  | 0.142  |
| | $pCO_2$  | 88 | 0.226  | 0.827  |
| | $pCO_2$ x Temperature  | 88 | -0.521  | 0.617  |
| Mean picocyanobacteria abundance ($\times 10^6$ cells L$^{-1}$) | Temperature | 88 | **5.983**  | **<0,001*** **  |
| | $pCO_2$ (10°C) | 44 | 1.480  | 0.213  |
| | $pCO_2$ (15°C) | 44 | **-3.051**  | **0.038***  |

| Measurement | Predictor | n | Estimate | p |
|---|---|---|---|---|
| | Temperature  | 88 | -0.015  | 0.988  |
| Particulate primary production (µmol C L⁻¹) | $pCO_2$  (10 °C) | 84 | -0.940  | 0.375  |
| | $pCO_2$ x Temperature  | 84 | 0.460  | 0.658  |
| | Temperature  | 88 | 1.894  | 0.095  |
| Dissolved primary production (µmol C L⁻¹) | $pCO_2$  | 88 | -1.145  | 0.285  |
| | $pCO_2$ x Temperature  | 88 | 0.847  | 0.422  |
| (Log) Chl *a*-normalized particulate primary production (µmol C (µg Chl *a*)⁻¹ d⁻¹)  | Temperature  | 88 | -2.288  | 0.052  |
| | $pCO_2$  (10 °C) | 84 | -1.491  | 0.174  |
| | $pCO_2$ x Temperature  | 84 | 1.105  | 0.301  |
| (Log) Chl *a*-normalized dissolved primary production (µmol C (µg Chl *a*)⁻¹ d⁻¹)  | Temperature  | 88 | **2.357**  | **0.046***  |
| | $pCO_2$ (10°C)  | 48 | -2.573  | 0.062  |
| | $pCO_2$ (15°C)  | 48 | 1.345  | 0.250  |

**Table 54. Results of the generalized least squares models (gls) tests for the effects of temperature, pCO₂ and their interaction.**
**Separate analysis with pCO₂ as a continuous factor were performed when temperature had a significant effect. Maximum particulate**
**and dissolved primary production, and time-integration over the full duration of the experiment (day 0 to day 13). Natural logarithm**
**transformation is indicated in parentheses when necessary, significant results are in bold. \*p < 0.05, \*\*p < 0.01.**

| Response Variable | Factor | dDf | t-value | p-value |
|---|---|---|---|---|
| Maximum particulate primary production ($\mu$mol C L$^{-1}$ d$^{-1}$) | Temperature | 8 | 2.466 | 0.039* |
| | pCO$_2$ (10 °C) | 4 | -2.328 | 0.080 |
| | pCO$_2$ (15 °C) | 4 | -2.394 | 0.075 |
| Time-integrated particulate primary production ($\mu$mol C L$^{-1}$ d$^{-1}$) | Temperature | 8 | -0.055 | 0.958 |
| | pCO$_2$ (10 °C) | 4 | -1.300 | 0.230 |
| | pCO$_2$ (15 °C) | 4 | 0.801 | 0.446 |
| (Log) Maximum dissolved primary production ($\mu$mol C L$^{-1}$) | Temperature | 8 | -0.659 | 0.528 |
| | pCO$_2$ | 8 | -3.342 | 0.010** |
| | pCO$_2 \times$ Temperature | 8 | 2.858 | 0.021* |
| Time-integrated dissolved primary production ($\mu$mol C L$^{-1}$) | Temperature | 8 | 1.687 | 0.130 |
| | pCO$_2$ | 8 | -2.153 | 0.063 |
| | pCO$_2 \times$ Temperature | 8 | 1.880 | 0.097 |

[Figure]

Formatted Table